# SEPARATION POWER OF EQUIVARIANT NEURAL NETWORKS

**Marco Pacini**[1,2]    **Xiaowen Dong**[3]    **Bruno Lepri**[2]    **Gabriele Santin**[4]
[1]University of Trento, [2]Fondazione Bruno Kessler,
[3]University of Oxford, [4]University of Venice

## ABSTRACT

The separation power of a machine learning model refers to its ability to distinguish between different inputs and is often used as a proxy for its expressivity. Indeed, knowing the separation power of a family of models is a necessary condition to obtain fine-grained universality results. In this paper, we analyze the separation power of equivariant neural networks, such as convolutional and permutation-invariant networks. We first present a complete characterization of inputs indistinguishable by models derived by a given architecture. From this results, we derive how separability is influenced by hyperparameters and architectural choices—such as activation functions, depth, hidden layer width, and representation types. Notably, all non-polynomial activations, including ReLU and sigmoid, are equivalent in expressivity and reach maximum separation power. Depth improves separation power up to a threshold, after which further increases have no effect. Adding invariant features to hidden representations does not impact separation power. Finally, block decomposition of hidden representations affects separability, with minimal components forming a hierarchy in separation power that provides a straightforward method for comparing the separation power of models.

## 1   INTRODUCTION

Alongside the proliferation and success of equivariant models (Wood & Shawe-Taylor, 1996; Cohen, 2021; Maron et al., 2018), there has been a growing interest in understanding the fundamental reasons behind their performances, and in assessing their expressive power (Maron et al., 2019a; Yarotsky, 2018). For traditional deep learning approaches, this expressive power is usually quantified in terms of universality (E, 2019), or their ability to approximate any element of a given class of functions to arbitrary precision. However, universality is not directly applicable to neural networks that incorporates invariances of the data (Bronstein et al., 2021), since they necessarily act by identifying pairs of inputs that are equivalent under the given set of transformations. This feature creates a complex interaction between the network's ability to discriminate different input data, and the invariant or equivariant structure that they are trying to preserve. Assessing expressivity thus requires first a fine-grained analysis of the separation power of these families of neural networks, namely their capacity of distinguishing distinct inputs, which is a necessary condition for the universality of the models (Joshi et al., 2023).

In the graph learning community, which is a paramount domain where invariant and equivariant models are studied (Maron et al., 2018; Puny et al., 2023; Bevilacqua et al., 2022), networks are required to be invariant or equivariant under the group of permutations of the graph's nodes. In this domain, the primary methods for comparing separation power are the Weisfeiler-Leman (WL) isomorphism test (Weisfeiler & Leman, 1968) and homomorphism counting (Lovász, 2012). Significant attention has been devoted to studying this property for graph learning models such as Graph Neural Networks (GNNs) (Scarselli et al., 2009; Gori et al., 2005; Kipf & Welling, 2017), Invariant Graph Networks (IGNs) (Maron et al., 2018; 2020), and subgraph GNNs (Alsentzer et al., 2020; Bevilacqua et al., 2022). However, the WL test and homomorphism counting, along with their variants, have severe limitations imposed by their combinatorial nature. In particular, recent research (Joshi et al., 2023) has highlighted the necessity of developing expressivity measures applicable to models that process data beyond relational structures, such as geometric graphs.

In this paper we contribute to this effort by studying the separation power of a more general class of equivariant neural networks, which are not covered by previous results but are of significant practical interest. Namely, we focus on the family of neural networks with regular convolutions (Cohen & Welling, 2016a), i.e., networks with non-polynomial continuous point-wise activations, finite-dimensional representations, and equivariance with respect to the action of finite groups acting on representations as permutations. This class is rich enough to comprise many models of common interest, such as IGNs (Maron et al., 2018), Circular Convolutional Neural Networks (Circular CNNs) (Ravanbakhsh, 2020), and Icosahedral CNNs (Cohen et al., 2019), even if the proposed approach is not able to address some relevant equivariant models such as Clebsh-Gordan or polynomial approaches (Kondor et al., 2018; Puny et al., 2023).

Specifically, we precisely describe the set of input pairs identified by relevant families of neural networks. In contrast, other approaches, limited to IGNs and graph processing, provide only upper bounds on expressiveness (Geerts, 2020) or lower bounds that require networks with large hidden feature widths (Maron et al., 2019a). Additionally, we show how hyperparameter and architectural choices impact the separation power of equivariant neural network models, both in general settings and in specific cases of practical interest.

To study the separation power of relevant classes of equivariant networks, we show that the set of identified points corresponds to the set of common zeros of a modified set of networks (Section 5.1). We characterize the set of input pairs identified by these families of neural networks by introducing an explicit formula which, remarkably, is recursive over the networks depth (Section 5.2). This result provides important insights into how different hyperparameters and architectural choices impact the design of practical equivariant neural network models. In particular, we prove that any non-polynomial activation is equivalent in separation power, achieving the maximum separability for networks with a fixed architecture (Section 5.3). We show that increasing depth enhances separation up to a certain depth, where separation power stabilizes (Section 5.4). Furthermore, we prove that the multiplicity of the blocks in hidden representations or, equivalently, the width of invariant hidden features does not affect the separation power of the networks (Section 5.5). We demonstrate that the separation power of different block types forms a hierarchy, corresponding to the partial ordering of sub-groups of the symmetry group with respect to which the model is equivariant (Section 5.6). We illustrate how these general results apply to practical models (Section 6). Specifically, we strengthen existing results by showing that a much broader class of IGNs matches the separation power of WL (Section 6.1). Then, we demonstrate that the separation power of circular CNNs depends on the filter size (Section 6.2).

All proofs are provided in the Appendix.

**Contributions.** Our contributions can be summarized as follows: **(i)** We address the separation power of equivariant neural networks by fully characterizing the set of points identified by networks with a fixed architecture (Proposition 2 and Theorem 1). **(ii)** We prove that any continuous, real, element-wise, non-polynomial activation is equivalent in separation power, achieving the maximum separability for networks with a fixed architecture (Theorem 2). **(iii)** We show that increasing depth enhances separation power up to a specific threshold, beyond which it stabilizes (Theorem 3). **(iv)** We illustrate how block decomposition of layers influences separability (Theorem 4) and how separation power is independent of invariant hidden features (Remark 1). Notably, this result implies that any $k$-IGN matches the separation power of $k$-WL, improving upon previous results that required IGNs to have large hidden feature widths Maron et al. (2019a). **(v)** Finally, we show that the minimal components from this decomposition form a hierarchy in separation power (Theorem 5).

## 2 RELATED WORK

Recently, equivariant deep learning models have gained popularity (Cohen & Welling, 2014; 2016a;b), being successful in diverse fields such as computer graphics (Qi et al., 2017), galaxy morphology prediction (Dieleman et al., 2015), computational biology (Joshi et al., 2024), and computational chemistry (Chanussot et al., 2021). In the case of permutation equivariance, often required in graph learning, the WL test has been adopted as the fundamental tool to measure the expressivity of GNNs (Xu et al., 2019; Morris et al., 2019) and has been used to derive upper bounds (Geerts et al., 2021) and lower bounds (Maron et al., 2019a) on the expressiveness of IGNs and GNNs (Geerts & Reutter,

2022). Recently, homomorphism counting has been proposed as a more fine-grained measure of expressivity for GNNs (Zhang et al., 2024), capable of assessing the separation power of subgraph GNNs and their variants (Alsentzer et al., 2020; Frasca et al., 2022; Bevilacqua et al., 2022). Other works that are related to the study of neural network separability include specific universality results for equivariant networks (Maron et al., 2019b; Yarotsky, 2018; Zhou, 2020; Ravanbakhsh, 2020; Keriven & Peyré, 2019; Dym & Maron, 2020). Instead, Joshi et al. (2023) addresses the problem of separability by generalizing the WL test from combinatorial structures to geometric graphs. In this paper, we shift from graph and relational domains to the continuous domain, where WL-based approaches are inapplicable, and extend the study of separation power to a broader class of equivariant neural networks not explored in previous research but essential for practical applications. Specifically, we examine neural networks utilizing regular $G$-convolutions (Cohen & Welling, 2016a). This class encompasses several widely used models, such as IGNs (Maron et al., 2018), Circular CNNs (Ravanbakhsh, 2020), and Icosahedral CNNs (Cohen et al., 2019).

## 3 THE RELEVANCE OF SEPARABILITY

### 3.1 SEPARATION-CONSTRAINED UNIVERSALITY

The universality property of neural networks enables them to approximate any continuous function with arbitrary precision, meaning there exists a sequence of networks that converges pointwise to each continuous function. Equivariant neural networks are designed to handle target functions with specific structures, represented by transformations that recognize equivalent inputs. However, this characteristic necessitates a deeper examination of their separation power. The separation power $\rho(\mathcal{N})$ of a subset $\mathcal{N} \subseteq \mathcal{C}(X, Y)$ of continuous functions between topological spaces $X$ and $Y$ is defined as follows.

**Definition 1.** *A function $f : X \to Y$ is said to* separate *two points $\alpha, \beta \in X$ if $f(\alpha) \neq f(\beta)$. A family of functions $\mathcal{N}$ from $X$ to $Y$ separates $\alpha, \beta \in X$ if there exists a function $f \in \mathcal{N}$ that separates $\alpha$ and $\beta$. If a function or a family of functions fails to separate two points, we say that it* identifies *them. The set of pairs of points that are identified by $\mathcal{N}$ define on an equivalence relation*

$$\rho(\mathcal{N}) = \{(\alpha, \beta) \in X \times X \mid f(\alpha) = f(\beta) \text{ for each } f \in \mathcal{N}\}. \tag{1}$$

When working with spaces of neural networks, which we refer to as *neural spaces* for brevity, their separation power transfers to the class of functions they can approximate, as shown by the following fact.

**Fact.** *Let $(f_n)_{n \in \mathbb{N}}$ be a sequence of functions in $\mathcal{N}$ that converges pointwise to $f$. If $\alpha, \beta \in X$ such that $f_n(\alpha) = f_n(\beta)$ for all $n \in \mathbb{N}$, then $f(\alpha) = f(\beta)$.*

In particular, $\mathcal{N}$ cannot approximate with arbitrary precision functions beyond ones respecting $\rho(\mathcal{N})$, namely, $\mathcal{C}_{\rho(\mathcal{N})}(X, Y) = \{f \in \mathcal{C}(X, Y) \mid f(\alpha) = f(\beta) \, \forall (\alpha, \beta) \in \rho(\mathcal{N})\}$. Understanding however if the entire set $\mathcal{C}_{\rho(\mathcal{N})}(X, Y)$ can be approximated leads to the study of *separation-constrained universality*.

Notably, Maron et al. (2019b) and Ravanbakhsh (2020) illustrate this phenomenon in the context of equivariant neural networks, which are proven to approximate any continuous equivariant function. However, their constructions involve intermediate representations of impractically large dimensions. In contrast, Geerts (2020) and Maron et al. (2019a) show that permutation equivariant networks commonly used in practice can approximate continuous permutation equivariant functions whose separation power is equivalent to the WL test.

In this work, *we address the problem of characterizing $\rho(\mathcal{N})$ for relevant families of equivariant neural networks*, as it is *necessary*, though not sufficient, to understand separation-constrained universality. Specifically, we focus on how hyperparameter and architecture choices influence separability, as we will discuss in Section 3.2.

### 3.2 THE EFFECT OF HYPERPARAMETERS ON SEPARABILITY AND UNIVERSALITY

From a practical viewpoint it is fundamental to understand the hyperparameters and architecture choices that affect the separation or approximation power of families of neural networks. For example,

IGN's separation and approximation power are influenced by two hyperparameters $(k, w)$, where $k$ represents the network's relational order and $w$ denotes the width of the final multi-layer perceptron. Informally, Maron et al. (2019b) showed that IGN $= \cup_{k,w} k\text{-IGN}_w$ is universal for continuous equivariant functions, while Geerts (2020) proved that $k\text{-IGN} = \cup_w k\text{-IGN}_w$ is universal only within the class of equivariant functions in $\mathcal{C}_{k\text{-WL}}(X, Y)$, the set of continuous equivariant functions with the same separation power as $k$-WL. This example highlights that, in a general equivariant setting, two types of hyperparameters and architecture choices may exist: **(i)** those like $k$, which regulate the separation power and, hence, have a huge impact on approximation power, but also have a significant impact on the required computational resources, and **(ii)** hyperparameters like $w$, which do not affect separability but may impact separation-constrained approximation, often with a limited impact on computational resources. In our work, we aim to identify which hyperparameters and architecture choices control separation power, determining which belong to the first category and which may fall into the second.

## 4 PRELIMINARIES

### 4.1 GROUPS AND EQUIVARIANCE

We aim to define functions that are symmetric with respect to specific transformations. Groups, which are particularly useful for computation and technical manipulation, consist of transformations that fulfill certain criteria: the elements can be combined, each element has an inverse, and there is a neutral element with respect to composition. While group theory efficiently studies symmetries and transformations from a purely algebraic perspective, it needs to be adapted to the linear algebra framework required for defining neural networks. Representation theory acts as a dictionary for translating between these two languages, showing how abstract groups can be mapped to sets of matrices that themselves form groups. For an overview, see Appendix A or Fulton & Harris (2004) for a more complete reference. We will primarily focus on permutation representations. Before defining them, we need to introduce additional notation. Let $X$ be a finite set and $G$ a finite group acting on it. Let $\mathbb{R}^X$ denote the set of real-valued functions defined on $X$. For each $x \in X$, let $e_x \in \mathbb{R}^X$ be the function that takes the value 1 at $x$ and 0 everywhere else, and note that the set $\{e_x\}_{x \in X}$ forms a basis for $\mathbb{R}^X$. A *permutation representation* of $G$ is a linear action of $G$ on $V = \mathbb{R}^X$ such that for each $g \in G$ and $x \in X$, we have $g(e_x) = e_{gx}$. Letting $V$ and $W$ be permutation representations of $G$, we say that a function $\phi : V \to W$ is $G$-equivariant if $\phi(gv) = g\phi(v)$ for each $v \in V$ and $g \in G$. We denote by $\mathrm{Hom}(V, W)$ the set of linear maps between $V$ and $W$, and by $\mathrm{Hom}_G(V, W)$ the subset of $G$-equivariant linear maps. Similarly, we refer to the set of affine maps between $V$ and $W$ as $\mathrm{Aff}(V, W)$, and the set of $G$-equivariant affine maps as $\mathrm{Aff}_G(V, W)$. Note that $\mathrm{Hom}(V, W)$, $\mathrm{Aff}(V, W)$, and their equivariant counterparts are real vector spaces with respect to addition and scalar multiplication. Moreover, Pacini et al. (2024) prove that a map $f \in \mathrm{Aff}(V, W)$ can be uniquely decomposed as $\tau_v \circ \phi$ for some $v \in V$ and $\phi \in \mathrm{Hom}(V, W)$ and it is equivariant if and only if its linear part $\phi$ is equivariant and $v \in W^G = \{v \in W \mid gv = v \ \forall g \in G\}$, the set of $G$-invariant vectors in $W$. Thus, we have relevant linear morphisms $\lambda : \mathrm{Aff}_G(V, W) \to \mathrm{Hom}_G(V, W)$, which projects an affine map onto its linear part, and $\tau : \mathrm{Aff}_G(V, W) \to W^G$, which projects an affine map onto its translational part.

### 4.2 EQUIVARIANT NEURAL NETWORKS

With all the necessary definitions in place, we can now introduce the notion of equivariant neural network. Our study will focus on neural networks that are equivariant with respect to finite groups, have arbitrary point-wise continuous activation functions, and whose representations are permutation representations.

**Definition 2** (Point-wise Activation). *Let $\mathbb{R}^X$ be a permutation representation of a group $G$, and let $\sigma : \mathbb{R} \to \mathbb{R}$. We define the* point-wise activation *induced by $\sigma$ as the function $\tilde{\sigma} : \mathbb{R}^X \to \mathbb{R}^X$ such that $\tilde{\sigma}(\sum_{x \in X} \alpha_x e_x) = \sum_{x \in X} \sigma(\alpha_x) e_x$. We will often abuse notation and refer to $\sigma$ as the activation function as well.*

**Definition 3** (Neural Networks and Neural Spaces). *Let $G$ be a group and $V_0, \ldots, V_d$ be permutation representations of $G$, and let $M_i$ be subsets of $\mathrm{Aff}_G(V_{i-1}, V_i)$ for $i = 1, \ldots, d$. Given $d \geq 2$, the*

neural space *with layers in* $M_1, \ldots, M_d$ *and activation* $\sigma$ *is the recursively defined set*

$$\mathcal{N}_\sigma(M_1, \ldots, M_d) = \left\{ \phi^d \circ \tilde{\sigma} \circ \eta^{d-1} \mid \phi^d \in M_d, \, \eta^{d-1} \in \mathcal{N}_\sigma(M_1, \ldots, M_{d-1}) \right\},$$

*where* $\mathcal{N}_\sigma(M_1) = M_1$. *A* neural network *with layers* $M_1, \ldots, M_d$ *and activation* $\sigma$ *is an element* $\eta^d \in \mathcal{N}_\sigma(M_1, \ldots, M_d)$. *When* $M_i = \mathrm{Aff}_G(V_{i-1}, V_i)$ *for each* $i = 1, \ldots, d$, *we simply write* $\mathcal{N}_\sigma(V_0, \ldots, V_d)$ *instead of* $\mathcal{N}_\sigma(M_1, \ldots, M_d)$.

In Definition 3, we do not impose any additional structure on $M_i$ beyond it being a subset of $\mathrm{Aff}_G(V_{i-1}, V_i)$, although we will primarily consider $M_i$ to be a vector subspace.

We now provide examples of relevant models that align with Definition 3.

**Example 1** (Equivariant Neural Networks)**.** *Let* $G$ *be a group,* $V_i = \mathbb{R}^{X_i}$ *be permutation representations of* $G$, *and* $M_i = \mathrm{Aff}_G(V_{i-1}, V_i)$ *the full space of equivariant affine maps from* $V_{i-1}$ *to* $V_i$ *for each* $i = 1, \ldots, d$. *The neural space* $\mathcal{N}_\sigma(M_1, \ldots, M_d)$, *or equivalently* $\mathcal{N}_\sigma(V_0, \ldots, V_d)$, *is the usual space of equivariant neural networks with depth* $d$ *and representation spaces* $V_i$ *for each* $i = 0, \ldots, d$.

As discussed in Section 4.1, equivariant affine maps $\mathrm{Aff}_G(V, W)$ can be decomposed into a linear part in $\mathrm{Hom}_G(V, W)$ and a translational part in the invariant subspace $W^G$, the set of $G$-invariant vectors of $W$. In our setting, the symmetry group $G$ is finite, and $W$ is a permutation representation, with its invariant part $W^G$ characterized by the following result.

**Proposition 1.** *Let* $\mathbb{R}^X$ *be a permutation representation of* $G$ *with orbit decomposition* $X_1 \sqcup \cdots \sqcup X_n$ *(see Definition 7 in Appendix A.3), let* $Y \subseteq X$. *Define* $\mathbb{1}_Y = \sum_{y \in Y} e_y \in \mathbb{R}^X$. *The invariant subspace of* $\mathbb{R}^X = \mathbb{R}^{X_1} \oplus \cdots \oplus \mathbb{R}^{X_n}$, *consisting of vectors fixed by the action of* $G$, *is generated by the basis* $\mathbb{1}_{X_1}, \ldots, \mathbb{1}_{X_n}$.

With this further characterization, we present detailed examples of equivariant affine maps and neural spaces relevant to machine learning applications, showing how common models can be expressed within this formalism. Furthermore, we will revisit these neural spaces in Section 6, highlighting specific properties related to their separation power.

**Example 2** (Invariant Graph Networks)**.** *Invariant Graph Networks of order 2 (2-IGNs) (Maron et al., 2018) are neural network models that ensure equivariance with respect to node ordering, making them particularly effective for graph processing tasks. They process graphs encoded as adjacency matrices* $A$ *in* $\mathbb{R}^{n \times n \times f}$, *where the first two dimensions encode the relational structure, and the third dimension corresponds to the features. In our setting, these matrices are defined as elements in* $\mathbb{R}^X \otimes \mathbb{R}^f$ *where* $X = [n] \times [n]$ *and* $\mathbb{R}^f$ *is the invariant feature space. Each element* $\sigma \in G = S_n$ *acts on* $X$ *as* $\sigma(i, j) = (\sigma(i), \sigma(j))$ *for each* $(i, j) \in X$ *and trivially on* $\mathbb{R}^f$. *To achieve permutation equivariance, the layers of 2-IGNs are maps in* $\mathrm{Aff}_{S_n}(\mathbb{R}^X \otimes \mathbb{R}^{f_{i-1}}, \mathbb{R}^X \otimes \mathbb{R}^{f_i})$, *and thus their associated neural spaces are* $\mathcal{N}_\sigma(\mathbb{R}^X \otimes \mathbb{R}^{f_0}, \ldots, \mathbb{R}^X \otimes \mathbb{R}^{f_d})$. *Proposition 10 in Appendix A.4 implies that understanding the structure of* $\mathrm{Aff}_{S_n}(\mathbb{R}^X \otimes \mathbb{R}^{f_{i-1}}, \mathbb{R}^X \otimes \mathbb{R}^{f_i})$ *reduces to understanding the structure of* $\mathrm{Aff}_{S_n}(\mathbb{R}^X, \mathbb{R}^X)$, *which is completely characterized in Maron et al. (2018) and Pearce-Crump (2023), which states that* $\dim \mathrm{Hom}_{S_n}(\mathbb{R}^X, \mathbb{R}^X) = 15$ *for* $n > 3$ *and, in accordance with Proposition 1, bias terms can be described by noting that* $X = [n] \times [n]$ *splits into two orbits,* $X_1 = \{(i, i) \mid i \in [n]\}$ *and* $X_2 = \{(i, j) \mid i \neq j\}$. *Identifying* $\mathbb{R}^{[n] \times [n]} = \mathbb{R}^n \otimes \mathbb{R}^n \cong \mathbb{R}^{n \times n}$, *elements in* $\mathbb{R}^{X_1}$ *correspond to diagonal matrices in* $\mathbb{R}^{n \times n}$, *while elements in* $\mathbb{R}^{X_2}$ *are off-diagonal matrices. In particular, invariant vectors in* $\mathbb{R}^{X_1}$ *are linear combinations of*

$$\mathbb{1}_{X_1} = \begin{bmatrix} 1 & 0 & 0 \\ 0 & 1 & 0 \\ 0 & 0 & 1 \end{bmatrix} \quad and \quad \mathbb{1}_{X_2} = \begin{bmatrix} 0 & 1 & 1 \\ 1 & 0 & 1 \\ 1 & 1 & 0 \end{bmatrix}.$$

*Then,*

$$\mathrm{Aff}_{S_n}(\mathbb{R}^X, \mathbb{R}^X) = \left\{ A \mapsto \sum_{i=1}^{15} x_i \phi^i(A) + y_1 \mathbb{1}_{X_1} + y_2 \mathbb{1}_{X_2} \mid x_1, \ldots, x_{15}, y_1, y_2 \in \mathbb{R} \right\},$$

*where* $\phi^1, \ldots, \phi^{15}$ *forms a basis of* $\mathrm{Hom}_{S_n}(\mathbb{R}^X, \mathbb{R}^X)$. *Furthermore, Maron et al. (2019b) show that 2-IGNs can be generalized to* $k$-*IGNs, which employ hidden representation of high order, namely elements in* $\mathbb{R}^{[n]^k} \otimes \mathbb{R}^{f_i}$.

**Example 3** (Circular Convolutional Neural Networks). *Circular convolutional filters can be described in the context of permutation representations, as we will demonstrate using the 1-dimensional case for simplicity. Let $X = [n]$ with the cyclic group $G = \mathbb{Z}_n$ acting on it in the standard way, and identify $\mathbb{R}^{[n]} = \mathbb{R}^n$. Linear maps in $\mathrm{Hom}_{\mathbb{Z}_n}(\mathbb{R}^n, \mathbb{R}^n)$ are circulant matrices $C(x)$ associated with a vector $x = (x_1, \ldots, x_n) \in \mathbb{R}^n$ (Davis, 1979). Hence, the linear part and the translational part of a map in $\mathrm{Aff}_{\mathbb{Z}_n}(\mathbb{R}^n, \mathbb{R}^n)$ are written as follows:*

$$C(x) = \begin{bmatrix} x_1 & x_n & x_{n-1} & \cdots & x_2 \\ x_2 & x_1 & x_n & \cdots & x_3 \\ x_3 & x_2 & x_1 & \cdots & x_4 \\ \vdots & \vdots & \vdots & \ddots & \vdots \\ x_n & x_{n-1} & x_{n-2} & \cdots & x_1 \end{bmatrix} \text{ and } y\mathbb{1}_X = y\mathbb{1}_{[n]} = y \begin{bmatrix} 1 \\ \vdots \\ 1 \end{bmatrix}.$$

*Note that $C(x) = \sum_{i=1}^{n} x_i C(e_i)$, where $e_1, \ldots, e_n$ is the canonical basis of $\mathbb{R}^n$. Therefore, since convolutional filters with limited support are more commonly used in practice, it is appropriate to restrict our choice of layers to maps within*

$$M^k = \left\{ v \mapsto \sum_{i=1}^{k} x_i C(e_i) v + y\mathbb{1}_{[n]} \mid x_1, \ldots, x_k, y \in \mathbb{R} \right\},$$

*which are the 1-dimensional counterpart of the $k \times k$ 2D convolutional filters widely used in computer vision. In this case, the corresponding neural space will be $\mathcal{N}_\sigma(M^{k_1}, \ldots, M^{k_d})$ for appropriate choices of filter sizes $1 \leq k_1, \ldots, k_d \leq n$.*

To generalize previous examples, we now assume that the layer spaces $M$, which are subspaces of $\mathrm{Aff}_G(V, \mathbb{R}^X)$, can be written as

$$M = \left\{ v \mapsto \sum_{i=1}^{k} x_i \phi^i(v) + \sum_{P \in \mathcal{P}} y_P \mathbb{1}_P \mid x_1, \ldots, x_k, y_P \in \mathbb{R} \text{ for all } P \in \mathcal{P} \right\},$$

where $\phi^1, \ldots, \phi^k$ generate a subspace of $\mathrm{Hom}_G(V, \mathbb{R}^X)$, and $\mathcal{P}$ is a partition of $X$, that may either combine several orbits from the orbit partition $X = X_1 \sqcup \cdots \sqcup X_n$ into larger subsets, or coincide with the orbit partition itself. For further technical details, we refer the interested reader to Definition 12 in Appendix B.1.

Now that we have described the structure of layer spaces in detail and explained why neural spaces constructed from these layer spaces more accurately reflect specific architectures used in practice, we can proceed to study the separation power of these neural spaces.

## 5 MAIN RESULTS

We begin by describing and formulating the twin network trick in Section 5.1, which serves as the primary tool for converting a separation problem into a zero locus problem, to be addressed informally in Section 5.2. In the subsequent sections, we will explore the implications of this result and how it can be applied to effectively compare the separation power of different neural spaces.

### 5.1 THE TWIN NETWORK TRICK

In this section, we introduce the twin network trick, which transforms a network separation problem into a zero locus problem for neural networks. This allows us to apply the recursive techniques for solving zero locus problems developed in Section 5.2. Specifically, a zero locus problem involves identifying all points that are mapped to zero by all networks within a given neural space.

More precisely, the identification equivalence relation in (1) can be reformulated as the following zero locus problem: $(\alpha, \beta) \in \rho(\mathcal{N}_\sigma(M_1, \ldots, M_d))$ if and only if

$$\eta(\alpha) - \eta(\beta) = 0 \ \ \forall \eta \in \mathcal{N}_\sigma(M_1, \ldots, M_d). \tag{2}$$

We observe that (2) reduces to a zero locus problem involving *twin networks*. The twin network $\overline{\eta} : V \oplus V \to W$, where $\overline{\eta}(\alpha, \beta) = \eta(\alpha) - \eta(\beta)$, is itself a neural network with the same depth as $\eta$ but with a different architecture. Namely, $\overline{\eta} \in \mathcal{N}_\sigma(\overline{M}_1, \ldots, \overline{M}_{d-1}, M'_d)$ where we define $M'_d = \{(\alpha, \beta) \mapsto \phi(\alpha) - \phi(\beta) \mid \phi \in M_d\}$ and
$$\overline{M}_i = \left\{ \overline{\phi} : (\alpha, \beta) \mapsto (\phi(\alpha), \phi(\beta)) \mid \phi \in M_i \right\}.$$
Thanks to the definition of the twin network, we can restate the identification problem in Equation 2 as an equivalent zero locus problem, where we aim to find all $\overline{\beta}$ in $V \oplus V$ that satisfy

$$\overline{\eta}(\overline{\beta}) = 0 \ \forall \overline{\eta} \in \mathcal{N}_\sigma(\overline{M}_1, \ldots, \overline{M}_{d-1}, M'_d).$$

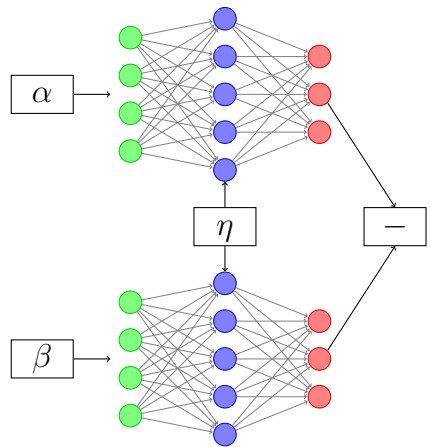

Figure 1: The twin network trick illustrated. Evaluating two copies of $\eta$ on $\alpha$ and $\beta$, and subtracting the resulting outputs, is equivalent to evaluating the twin network $\overline{\eta}$ on $(\alpha, \beta)$.

In summary, these observations can be synthesized into the following proposition, which directly links the identification relation to a zero locus.

**Proposition 2.** *For a family $\mathcal{F}$ of functions between a set $V$ and a vector space $W$, let*
$$\mathcal{I}(\mathcal{F}) = \{\beta \in V \mid \eta(\beta) = 0 \ \forall \eta \in \mathcal{F}\}.$$
*be the* zero locus *of $\mathcal{F}$. Then, for any neural space $\mathcal{N}_\sigma(M_1, \ldots, M_d)$, we have*
$$\rho(\mathcal{N}_\sigma(M_1, \ldots, M_d)) = \mathcal{I}(\mathcal{N}_\sigma(\overline{M}_1, \ldots, \overline{M}_{d-1}, M'_d)).$$

Our task, then, is to determine the zero locus corresponding to the neural space of twin networks.

## 5.2 The Characterization Theorem

In the previous sections, thanks to Proposition 2, we have translated the problem of computing the identification relation $\rho(\mathcal{N}_\sigma(M_1, \ldots, M_d))$ into the problem of computing the zero locus $\mathcal{I}(\mathcal{N}_\sigma(\overline{M}_1, \ldots, \overline{M}_{d-1}, M'_d))$. More generally, this zero locus can be determined using the recursive formula proposed in Theorem 1. For brevity, we provide an informal version here, and refer the interested reader to Theorem 7 in Appendix B.1 for the complete version.

We begin by recalling and defining the necessary notation to state Theorem 1. Let $M_i$ be vector subspaces of $\mathrm{Aff}_G(\mathbb{R}^{X_{i-1}}, \mathbb{R}^{X_i})$ for $i = 1, \ldots, d$. Recall that $\lambda(M_d)$ denotes the linear part of $M_d$, and let $\phi^{d,1}, \ldots, \phi^{d,s_d}$ be linear maps spanning $\lambda(M_d)$, and recall that $\tau(M_d) = \langle \mathbb{1}_P \rangle_{P \in \mathcal{P}}$ for some partition $\mathcal{P}$ of $X_d$. Let $\mathcal{Q}$ be another partition of $X_d$; if $\mathcal{Q}$ is finer than $\mathcal{P}$ we indicate this relationship as $\mathcal{Q} \leq \mathcal{P}$. Furthermore, for each $h = 1, \ldots, s_d$ and $k \in X_d$ define the family of partitions of $X_d$

$$\Psi_{h,k} = \left\{ \mathcal{Q} \leq \mathcal{P} \mid \sum_{i \in P} \phi^{d,h}_{ki} = 0, \ \forall P \in \mathcal{Q} \right\}.$$

Let $\pi_i : \mathbb{R}^{X_{d-1}} \to \mathbb{R}$ denote the projection onto the $i$-th component of $\mathbb{R}^{X_{d-1}}$ for each $i$ in $X_{d-1}$. For each $i, j \in X_{d-1}$, define the set $(M_{d-1})_{ij} = \{\phi' : x \mapsto \pi_i \phi(x) - \pi_j \phi(x) \mid \phi \in M\}$ which represents scalar-valued layers obtained as the differences between the $i$-th and $j$-th part of the $(d-1)$-th layer.

**Theorem 1** (Informal). *Using the notation defined above, if $\sigma$ is a non-polynomial continuous activation function, the following formula, recursive with respect to the depth $d$, holds*

$$\mathcal{I}(\mathcal{N}_\sigma(M_1, \ldots, M_d)) = \bigcap_{h,k} \bigcup_{\mathcal{Q} \in \Psi_{h,k}} \bigcap_{\substack{P \in \mathcal{Q} \\ i,j \in P}} \mathcal{I}(\mathcal{N}_\sigma(M_1, \ldots, M_{d-2}, (M_{d-1})_{ij})). \tag{3}$$

The theorem shows that the zero locus of a neural space of depth $d$ can be recursively computed as a combination of unions and intersections of the zero loci of neural spaces of depth $d - 1$. At depth 1, the neural space reduces to a subspace of affine maps, and finding its zero locus corresponds to solving a system of linear equations. Although the actual execution of Formula 3 requires superpolynomial time, this recursive approach is particularly useful for deriving key properties of the identification relation, such as the role of activations, depth, and hidden features on the separation power, as detailed in the following sections.

## 5.3 THE ROLE OF ACTIVATIONS

The following result shows that the choice of the activation function–and its properties, such as injectivity or monotonicity–is irrelevant to separability, as long as the activation is non-polynomial.

**Theorem 2.** *Let $\sigma$ and $\tau$ be two continuous activation functions, with $\sigma$ being non-polynomial. Then*

$$\rho(\mathcal{N}_\sigma(M_1, \ldots, M_d)) \subseteq \rho(\mathcal{N}_\tau(M_1, \ldots, M_d)).$$

*If $\tau$ is also non-polynomial, equality holds. Thus, non-polynomial activations not only yield equivalent separability but also achieve maximal separation power.*

*Proof outline.* The inclusion follows by reconstructing the steps of in the proof of Theorem 1 and applying the first part of Theorem 9. To prove the equality in the case $\tau$ is also non-polynomial, we reduce to solve the equivalent zero-locus problem thanks to Proposition 2. We proceed by induction on $d$ to show that non-polynomial activation functions do not affect the zero-locus. For the base case $d = 1$, we have $\mathcal{I}(\mathcal{N}_\sigma(M_1)) = \mathcal{I}(M_1)$, which is independent of $\sigma$. Now, assume as the inductive hypothesis that $\mathcal{I}(\mathcal{N}_\sigma(M_1, \ldots, M_{d-1}))$ is independent of $\sigma$ for any sequence $M_1, \ldots, M_{d-1}$. Additionally, by definition, $h$, $k$ and $\Psi_{h,k}$ are also independent of $\sigma$, which proves that none of the terms in the right-hand side of (3) depend on $\sigma$, thereby completing the proof. $\square$

While non-polynomiality is sufficient for maximal separation, some polynomial activations may also achieve maximal separation power. Identifying these polynomials–or simply some of their properties, such as their degree–remains a complex mathematical problem. For more details, see Kiss & Laczkovich (2014).

## 5.4 THE ROLE OF DEPTH

Depth is a key hyperparameter influencing the separation power of neural spaces. Theorem 3 shows that, while adding layers of the same type can initially enhance separation power, this effect stabilizes after a finite number of layers.

**Theorem 3.** *Let $M_i$ be a subspace of $\mathrm{Aff}_G(V_{i-1}, V_i)$ for each $i = 1, \ldots, d$. Suppose that $V_{h-1} = V_h$ for some integer $1 \leq h \leq d$ and that $id_{V_h} \in M_h$. If $\sigma$ is a continuous non-polynomial activation function, then for $m \leq n$,*

$$\rho(\mathcal{N}_\sigma(M_1, \ldots, M_{h-1}, \underbrace{M_h, \ldots, M_h}_{n \text{ times}}, M_{h+1}, \ldots, M_d)) \subseteq$$

$$\subseteq \rho(\mathcal{N}_\sigma(M_1, \ldots, M_{h-1}, \underbrace{M_h, \ldots, M_h}_{m \text{ times}}, M_{h+1}, \ldots, M_d)),$$

*but there exists a repetition threshold $R$ such that for all $n, m \geq R$ the inclusion becomes an equality.*

*Proof outline.* To prove the first part of the statement, it suffices to show the inclusion for $n = 1$ and $m = 2$. Moreover, Theorem 2 implies that it is sufficient to prove this inclusion for a single non-polynomial $\sigma$. Therefore, let $\sigma$ be the ReLU activation function, noting that in this case $\sigma \circ \sigma = \sigma$; equivalently $\tilde{\sigma} = \tilde{\sigma} \circ \tilde{\sigma} = \tilde{\sigma} \circ id_{\mathbb{R}^{x_i}} \circ \tilde{\sigma}$. Thus, each neural network $\phi_d \circ \tilde{\sigma} \circ \cdots \circ \tilde{\sigma} \circ \phi_1$ in $\mathcal{N}_\sigma(M_1, \ldots, M_h, \ldots, M_d)$ can be written as $\phi_d \circ \cdots \circ \tilde{\sigma} \circ id_{\mathbb{R}^{x_i}} \circ \tilde{\sigma} \circ \cdots \circ \tilde{\sigma} \circ \phi_1$, which is an element of $\mathcal{N}_\sigma(M_1, \ldots, M_h, M_h, \ldots, M_d)$, thereby proving the desired inclusion by applying Lemma 5. The stabilization property can be proved by reducing to the equivalent zero-locus problem and observing that descending sequences of finite intersections and unions of a finite number of sets, as given by Theorem 1, stabilize thanks to Lemma 2. $\square$

The repetition threshold may vary depending on the model and representation. For example, $k$-IGNs, being equivalent to $k$-WL, have a repetition threshold proportional to that of $k$-WL itself (Maron et al., 2019a; Geerts, 2020). In contrast, the following proposition shows an example of stabilization after just one repetition.

**Proposition 3.** *When the hidden representation spaces are regular representations, stabilization occurs after one layer repetition. Namely, $\rho(\mathcal{N}_\sigma(V, \mathbb{R}^G, \ldots, \mathbb{R}^G, \mathbb{R}^{G/H})) = \rho(\mathcal{N}_\sigma(V, \mathbb{R}^G, \mathbb{R}^{G/H}))$.*

## 5.5 THE ROLE OF INTERMEDIATE REPRESENTATIONS

In this section, we show that if a representation, $V$ can be decomposed as $V' \oplus V''$, then the separation power of neural spaces with hidden representation $V$ reduces to the combined separation power of two distinct neural spaces with hidden representations $V'$ and $V''$. In this section, we present Theorem 4, an additional application of Theorem 1, which demonstrates that the identification equivalence relation for neural spaces defined on $V$ is the intersection of those for neural spaces defined on $V'$ and $V''$. This implies that by decomposing each hidden representation V into a sum of *minimal* factors, the study of the separation power of general neural spaces can be reduced to analyzing those defined on minimal representations, as will be explored in Section 5.6.

**Theorem 4.** *Let $\sigma : \mathbb{R} \to \mathbb{R}$ be a continuous non-polynomial activation function, and let $V_0, \dots, V_d$ be permutation representations with $V_i = V_i' \oplus V_i''$ for some $0 \le i \le d$. Then,*

$$\rho(\mathcal{N}_\sigma(V_0, \dots, V_d)) = \rho(\mathcal{N}_\sigma(V_0, \dots, V_i', \dots, V_d)) \cap \rho(\mathcal{N}_\sigma(V_0, \dots, V_i'', \dots, V_d)).$$

*Remark* 1. Note that if $V_i' = V_i''$, we have

$$\rho(\mathcal{N}_\sigma(V_0, \dots, V_i' \oplus V_i'', \dots, V_d)) = \rho(\mathcal{N}_\sigma(V_0, \dots, V_i', \dots, V_d)). \tag{4}$$

As a result,

$$\rho(\mathcal{N}_\sigma(V_0, \dots, V_i \otimes \mathbb{R}^f, \dots, V_d)) = \rho(\mathcal{N}_\sigma(V_0, \dots, V_i, \dots, V_d)),$$

since $V_i \otimes \mathbb{R}^f \cong V_i^{\oplus f}$. Thus, the separability is independent of multiplicity and invariant features in intermediate representations.

## 5.6 THE ROLE OF REPRESENTATION TYPE

Thanks to Theorem 4, we can focus on studying the separation power of neural spaces defined on *minimal* representations. These minimal representations are of the form $\mathbb{R}^X$, where the group $G$ acts transitively on $X$. That is, for any pair of points $x, y \in X$, there exists an element $g \in G$ such that $gx = y$. Basic group theory (Fulton & Harris, 2004) shows that a set with a transitive action is in bijective correspondence with right cosets $G/H$ for some subgroup $H < G$, see Definition 6 in Appendix A.2. Informally, the following theorem allows us to compare representations induced by transitive actions arising from certain subgroups.

**Theorem 5.** *Let $K < H < G$ be finite groups. We have*

$$\rho(\mathcal{N}_\sigma(V, \dots, \mathbb{R}^{G/K}, \dots, W)) \subseteq \rho(\mathcal{N}_\sigma(V, \dots, \mathbb{R}^{G/H}, \dots, W)).$$

*Proof outline.* The proof consists in showing that $\mathbb{R}^{G/H}$ is a sub-representation of $\mathbb{R}^{G/K}$, and proving that this induces an embedding of $\mathcal{N}_\sigma(V, \dots, \mathbb{R}^{G/H}, \dots, W)$ into $\mathcal{N}_\sigma(V, \dots, \mathbb{R}^{G/K}, \dots, W)$. Consequently, the result follows directly from Lemma 5. $\qquad\square$

Theorem 5 implies that neural spaces with minimal representations in one layer, namely $\{\mathcal{N}_\sigma(V, \dots, \mathbb{R}^{G/H}, \dots, W)\}_{H < G}$, form a separation power hierarchy corresponding to the hierarchy of subgroups of $G$. In particular, if $H = G$, the corresponding representation $\mathbb{R}^{G/G}$ has minimal separation power. Furthermore, notice that $\mathbb{R}^{G/G} \cong \mathbb{R}$ is the trivial representation, and this means that invariant layers have the lowest separation power. On the other hand, if $H = \{e\}$ is the group containing only the identity element, the corresponding representation $\mathbb{R}^{G/\{e\}}$ has maximal separation power, since $\{e\}$ is contained in every subgroup of $G$. As $\mathbb{R}^{G/\{e\}} \cong \mathbb{R}^G$ is the regular representation, this implies that the regular representation achieves the maximum separation power. In general, if $K < H$, then $\dim \mathbb{R}^{G/H} < \dim \mathbb{R}^{G/K}$. Hence, improving separability requires working in a larger space, which, aside from ad-hoc optimizations, leads to additional computational cost. In particular, by applying Theorem 1, we can prove the following proposition.

**Proposition 4.** *The neural space $\mathcal{N}_\sigma(V, \mathbb{R}^G, \mathbb{R}^{G/H})$ of equivariant shallow networks with regular hidden representations identifies inputs if and only if they belong to the same $H$-orbit, i.e., $(\beta, \beta') \in \rho(\mathcal{N}_\sigma(V, \mathbb{R}^G, \mathbb{R}^{G/H}))$ if and only if there exists some $h \in H$ such that $h\beta = \beta'$.*

This is consistent with the results in Ravanbakhsh et al., which demonstrate that shallow networks with hidden representation blocks isomorphic to $\mathbb{R}^G$ are universal. This universality implies maximal separation power, as stated in Theorem 16 of Joshi et al. (2023).

## 6 IMPLICATIONS ON PRACTICAL MODELS

### 6.1 INVARIANT GRAPH NETWORKS

Theorem 1 in (Maron et al., 2019a) and Theorem 2 in (Geerts, 2020) together imply the following fundamental result for the theory of IGNs.

**Proposition 5.** *There exist $d > 0$ and a **large** $F > 0$ such that for hidden feature dimensions $f_1, \ldots, f_d > F$, the neural space $\mathcal{N}_\sigma((\mathbb{R}^n)^{\otimes 2} \otimes \mathbb{R}^{f_0}, (\mathbb{R}^n)^{\otimes k} \otimes \mathbb{R}^{f_1}, \ldots, (\mathbb{R}^n)^{\otimes k} \otimes \mathbb{R}^{f_d}, \mathbb{R})$ matches the separation power of $k$-WL.*

However, Remark 1 shows that the dimension of hidden invariant features does not affect separation power, strengthening Proposition 5 in the following corollary.

**Corollary 1.** *There exist $d > 0$ such that for **any** hidden feature dimensions $f_1, \ldots, f_d > 0$, the neural space $\mathcal{N}_\sigma((\mathbb{R}^n)^{\otimes 2} \otimes \mathbb{R}^{f_0}, (\mathbb{R}^n)^{\otimes k} \otimes \mathbb{R}^{f_1}, \ldots, (\mathbb{R}^n)^{\otimes k} \otimes \mathbb{R}^{f_d}, \mathbb{R})$ matches the separation power of $k$-WL.*

### 6.2 CONVOLUTIONAL NEURAL NETWORKS

The separation power of circular CNNs is influenced by the width of the filter's support.

**Proposition 6.** *Let $M^k$ be the layer space for circular convolutions with filter size $k$, as defined in Example 3. Consider the neural space $k\text{-CNN} = \mathcal{N}_\sigma(M^k, \mathrm{Aff}_{\mathbb{Z}_n}(\mathbb{R}^n, \mathbb{R}))$. This is the space associated with shallow convolutional networks, where the first layer consists of one filter of size $k$ followed by an output invariant layer. For $n > 2$, we have:*

$$\rho(n\text{-}CNN) \subsetneq \rho(1\text{-}CNN), \quad and \quad \rho(n\text{-}CNN) \subseteq \cdots \subseteq \rho(2\text{-}CNN) \subseteq \rho(1\text{-}CNN).$$

## 7 LIMITATIONS

The primary limitations of the proposed framework lie within its initial assumptions. Specifically, it only applies to permutation representations and cannot extend to other important equivariant models, such as Clebsch-Gordan or polynomial approaches (Kondor et al., 2018; Puny et al., 2023). While the techniques used to prove Theorem 1 could be applied, the functional equation (12) necessary for the proof is substantially different, and no actionable solution is known up to our knowledge. Solving these equations would improve our understanding of the separation power in these models. The second key assumption is that we consider only intermediate layers with bias, which is standard in many practical models. In cases where bias terms are absent, such as in some GNN models, our model can only provide a bound on the separation power. Indeed, these models can be reformulated as IGNs (Maron et al., 2018), which belong to a broader neural space with bias terms, allowing us to analyze their separation power. Our final assumption involves the use of non-polynomial activation functions, commonly employed with examples like $\mathrm{ReLU}$, $\tanh$, and sigmoid. While non-polynomiality is sufficient for separation, some polynomial activations may also achieve maximal separation power. However, identifying these specific polynomials presents a complex mathematical challenge. For further details, we refer readers to (Kiss & Laczkovich, 2014).

## 8 CONCLUSIONS

The proposed results enhance our understanding of which target functions can be approximated with arbitrary precision by functions in neural network spaces relevant to practical applications. In particular, these target functions must satisfy the identification relation of neural networks spaces, which can now be computed using Theorem 1. This result helps us classify hyperparameters and architectural choices into two categories: those that directly influence separation power, significantly impacting both approximation ability and computational cost, and those that do not affect separation power but may influence approximation in a separation-constrained setting, potentially with minimal computational cost. Specifically, we prove that all non-polynomial activations provide maximum separation power, depth enhances separation up to a stabilization point, and hidden feature width has no effect on separation power. Lastly, we show how these insights can be applied to practical models, such as IGNs and standard CNNs.

## 9 ACKNOLEDGEMENTS

Bruno Lepri acknowledges the support of the PNRR project FAIR - Future AI Research (PE00000013), under the NRRP MUR program funded by the NextGenerationEU and the support of the European Union's Horizon Europe research and innovation program under grant agreement No. 101120237 (ELIAS).

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

## A  PRELIMINARIES

### A.1  SPACES OF AFFINE TRANSFORMATIONS

Let $V$ and $W$ be vector spaces, we denote by $\mathrm{Hom}(V, W)$ the set of linear maps between $V$ and $W$, and $\mathrm{Aff}(V, W)$ the set of affine maps between $V$ and $W$. Note that both $\mathrm{Hom}(V, W)$ and $\mathrm{Aff}(V, W)$ are real vector spaces with respect to addition and scalar multiplication. For each $v \in W$ define $\tau_v : W \to W$ such as $\tau_v(w) = w + v$ for each $w$ in $W$. Each affine map $f : V \to W$ has a unique decomposition $\phi = \tau_v \circ \phi$, where $\phi$ is a linear map and $\tau_v$ is a translation by a vector $v \in W$. Previous observations imply that the map

$$\theta : \begin{array}{c} \mathrm{Hom}(V, W) \oplus W \to \mathrm{Aff}(V, W) \\ (\phi, v) \mapsto \tau_v \phi \end{array}$$

is an isomorphism of vector spaces. Define $T_W = \{\tau_v \mid v \in W\}$, as clearly $T_W \cong W$, we will often abuse notation identifying $T_W$ with $W$. Then, we can define the maps

$$\lambda : \begin{array}{c} \mathrm{Aff}(V, W) \to \mathrm{Hom}(V, W) \\ f \mapsto f - f(0) \end{array} \quad \text{and} \quad \tau : \begin{array}{c} \mathrm{Aff}(V, W) \to W \\ f \mapsto f(0). \end{array}$$

We call $\lambda(f)$ the *linear part* of $f$ for each $f \in \mathrm{Aff}(V, W)$ and $\tau(f)$ the *translational part* of $f$ for each affine map.

**Proposition 7.** *Maps $\lambda$ and $\tau$ are linear and surjective.*

*Proof.* The projection on the linear part $\lambda$ is linear since for each $\alpha, \beta \in \mathbb{R}$ and $f, g \in \mathrm{Aff}(V, W)$

$$\lambda(\alpha f + \beta g) = (\alpha f + \beta g) - (\alpha f + \beta g)(0) = \alpha f + \beta g - \alpha f(0) - \beta g(0) =$$
$$\alpha(f - f(0)) - \beta(g - g(0)) = \alpha \lambda(f) + \beta \lambda(g).$$

Furthermore, $\lambda$ is surjective since for each $\phi \in \mathrm{Hom}(V, W)$, we have that $\lambda(\phi) = \phi - \phi(0) = \phi$. The projection on the translational part $\tau$ is linear since for each $\alpha, \beta \in \mathbb{R}$ and $f, g \in \mathrm{Aff}(V, W)$

$$\tau(\alpha f + \beta g) = (\alpha f + \beta g)(0) = \alpha f(0) + \beta g(0) = \alpha \tau(f) + \beta \tau(g).$$

Map $\tau$ is also surjective since fix $f \in \mathrm{Hom}(V, W)$ and notice that $\tau(\tau_v f) = v$ and $\tau_v f \in \mathrm{Aff}(V, W)$ for each $v \in W$. $\qquad \square$

In general, we will be interested in vector sub-spaces $M$ of $\mathrm{Aff}(V, W)$. A simple way to construct $M$ is to identify sub-spaces of interest $L$ in $\mathrm{Hom}(V, W)$ and subspaces $T$ of $T_W \cong W$ and define $M = \theta(L \oplus T)$. We will be interested only is sub-spaces which are possible to construct in this way. Nevertheless, it is worth noticing there exist sub-spaces of $\mathrm{Aff}(V, W)$ that are not isomorphic through $\theta$ to directed sums of linear and translation parts. An example is given by the space $M = \{\tau_{\alpha v} \circ \alpha \phi \mid \alpha \in \mathbb{R}\}$. Notice that $\lambda(M) = \{\alpha \phi \mid \alpha \in \mathbb{R}\}$, $\tau(M) = \{\alpha v \mid \alpha \in \mathbb{R}\}$, and $\dim M = \lambda(M) = \tau(M) = 1$. If $M = \theta(\lambda(M) \oplus \tau(M))$, we should have $1 = \dim M = \dim \lambda(M) + \dim \tau(M) = 2$, which is not possible. Spaces that can be decomposed as $M = \theta(L \oplus T)$ have the following useful property.

**Proposition 8.** *Let $M$ be a subspace of $\mathrm{Aff}(V, W)$, where there exist a subspace $L$ of $\mathrm{Hom}(V, W)$ and a subspace $T$ of $W$ such that $M = \theta(L \oplus T)$. Let $\phi_1, \ldots, \phi_s$ be a basis for $L$ and $v_1, \ldots, v_n$ a basis for $T$. Then, each $f$ in $M$ can be written as*

$$f : \beta \mapsto \sum_{i=1}^{s} x_i \phi_i(\beta) + \sum_{j=1}^{n} y_j v_j,$$

*for unique parameters $x_i$ and $y_j$ for each $i = 1, \ldots, s$ and $j = 1, \ldots, n$.*

### A.2  GROUP THEORY

**Definition 4.** *A group is a pair $(G, \cdot)$ where $G$ is a set and $\cdot : G \times G \to G$ is a function satisfying the following axioms.*

- Associativity: *for each $g, h, k \in G$ we have $(g \cdot h) \cdot k = g \cdot (h \cdot k)$.*

- Identity: *there exists an element $e \in G$ such that $g \cdot e = e \cdot g = g$ for each $g \in G$.*

- Inverse Element: *for each element $g \in G$, there exists an element $g^{-1} \in G$ such that $g \cdot g^{-1} = g^{-1} \cdot g = e$.*

*A group is* finite *if it contains a finite number of elements. A group is* abelian *or* commutative *if $gh = hg$ for each $g, h \in G$.*

**Example 4.** *Here we present some fundamental examples of groups.*

- *Let $X$ be a set and define the set of permutation of $X$ as*

$$\mathcal{S}_X = \{f : X \to X \mid f \text{ is bijective}\}.$$

*With the composition operation form the* symmetric group *or the* permutation group *of $X$. Particular attention is devoted to the case $X = [n]$, we write $S_n = \mathcal{S}_X$ and it called* symmetric group *or the* permutation group *of $n$ elements.*

- *Let $\mathbb{Z}_n$ be the group of integers modulo $n$ with the addition operation, they are called* finite cyclic groups *of order $n$.*

- *Given two groups $G$ and $H$, the direct product $G \times H$ of them is still a group. The set of the elements is the Cartesian product of $G$ and $H$ while the sum is defined as*

$$(g_1, h_1) \cdot_{G \times H} (g_2, h_2) = (g_1 \cdot_G g_2, h_1 \cdot_H h_2).$$

Now, we introduce the notion of group homomorphism, a transformation between groups which preserves the operation.

**Definition 5.** *A* group homomorphism *is a map*

$$\phi : G \to H$$

*between $G$ and $H$ groups such that, for each $g, h \in G$*

$$\phi(g \cdot h) = \phi(g) \cdot \phi(h).$$

**Definition 6** (Cosets). *Let $G$ be a group and $H$ be a subgroup of $G$. The* set of left cosets *of $G$ by $H$ is the set $G/H = \{gH \mid g \in G\}$, where $gH = \{gh \mid h \in H\}$ are the* left cosets *of $H$. Similarly, we define the set of* right cosets *as $H \backslash G = \{Hg \mid g \in G\}$. Let $K$ be a second subgroup of $G$, we define the* double coset *of $H$ and $K$ with respect to an element $g \in G$ as the set $HgK = \{hgk \mid h \in H, k \in K\}$. The set of double cosets is denoted as $H \backslash G / K$.*

**Example 5.** *Relevant examples of left cosets include the following:*

1. *Consider $G = \mathbb{Z}$ and the subgroup $H = n\mathbb{Z}$ of integers multiples of $n$. The quotient $G/H$ is a group and is isomorphic to the cyclic group of $n$ elements, $\mathbb{Z}_n$.*

2. *Consider $G = S_3$, the symmetric group on three elements, and the subgroup $H = \{(1), (12)\}$. The quotient $G/H$ is a group and is isomorphic to $S_2$, symmetric group on two elements.*

3. *Consider $G = S_n$, the symmetric group on $n$ elements, and the subgroup $H = A_n$, the alternating group on $n$ elements. The quotient $G/H$ is a group and is isomorphic to $\mathbb{Z}_2$.*

## A.3 GROUP ACTIONS AND EQUIVARIANT MAPS

Let $G$ be a group and $X$ be a set. An *action* of the group $G$ on the set $X$ is a function

$$\Phi : G \times X \to X,$$

usually written as $\phi_g(x) = \Phi(g, x)$ for each $g$ in $G$ and $x$ in $X$, such that:

- For the identity element $e$ in $G$, the *identity condition* $\phi_e = id_X$ holds.
- For all $g, h \in G$, the *compatibility condition* $\phi_g \circ \phi_h = \phi_{gh}$ holds.

In this context, we often write $g \cdot x$ or simply $gx$ instead of $\phi_g(x)$. A $G$-set is a set $X$ equipped with a group action of $G$. This means that there is a well-defined action $\cdot : G \times X \to X$ satisfying the properties of a group action as described above. Throughout the following sections, it will often be convenient to decompose $G$-sets into a disjoint union of subsets, each minimal (in a sense specified in Definition 7) and equipped with a compatible $G$-action.

**Definition 7.** *Let $G$ be a group acting on a set $X$. An* orbit *in $X$ is a subset $Y \subseteq X$ such that for each $x \in Y$, we have $Y = \{g \cdot x \mid g \in G\}$. The set $X$ can be decomposed into a disjoint union of orbits under the action of $G$. This is called the* orbit decomposition *of $X$, and if $X$ is finite, the decomposition can be written as*

$$X = X_1 \sqcup \cdots \sqcup X_n,$$

*where $X_1, \ldots, X_n$ are the distinct orbits of $X$.*

Another fundamental concept for our treatment is that of a function between $G$-sets that preserves actions, which is more formally specified in Definition 8.

**Definition 8.** *Let $X$ and $Y$ be two $G$-sets, a map $f : X \to Y$ is $G$-equivariant if*

$$g \cdot f(x) = f(g \cdot x)$$

*for each $x$ in $X$.*

### A.4 Group Representations and Equivariant Affine Transformations

Let $G$ be a group and $V$ be a vector space over a field $\mathbb{R}$. A $G$-action $\Phi : G \times V \to V$ on $V$ is *$G$-representation* if $\phi_g$ is linear for each $g$ in $G$. Or equivalently,

$$\phi : \begin{array}{c} G \to \mathrm{GL}(V) \\ g \mapsto \phi_g \end{array}$$

where $\mathrm{GL}(V)$ is the general linear group of $V$, consisting of all invertible linear transformations on $V$. We will usually identify the entire $\Phi : G \times V \to V$ action with $V$ itself and write $gv = \Phi(g, v)$.

Let $V$ and $W$ be two $G$-representations, we will indicate the set of equivariant linear maps between $V$ and $W$ as $\mathrm{Hom}_G(V, W)$ and as $\mathrm{Aff}_G(V, W)$ the set of equivariant affine maps. Note that $\mathrm{Hom}_G(V, W)$ is a vector space. Indeed, $0 \in \mathrm{Hom}_G(V, W)$ and for each $f, g \in \mathrm{Hom}_G(V, W)$ and each $\alpha, \beta \in \mathbb{R}$, $\alpha f + \beta g \in \mathrm{Hom}_G(V, W)$. The same is true for $\mathrm{Aff}_G(V, W)$.

Let $V$ be a $G$-representation, we define the set of invariant vectors $V^G = \{v \in V \mid gv = v \ \forall g \in G\}$.

In Section A.1 we studied the properties of vector subspaces of affine transformations between vector spaces. Here we are now interested in studying subspaces of equivariant affine transformations. To better understand the structure of such spaces it is necessary to remind Theorem 8 in Pacini et al. (2024).

**Theorem 6.** *An map $\phi = \tau_w \circ f$ belongs to $\mathrm{Aff}_G(V, W)$ if and if $f \in \mathrm{Hom}_G(V, W)$ and $v$ is invariant.*

We can define restrictions of maps $\theta_G$, $\lambda_G$, and $\tau_G$ as follows

$$\theta_G : \begin{array}{c} \mathrm{Hom}_G(V, W) \oplus W^G \to \mathrm{Aff}_G(V, W) \\ (\phi, v) \mapsto \tau_v \phi \end{array}$$

$$\lambda_G : \begin{array}{c} \mathrm{Aff}_G(V, W) \to \mathrm{Hom}_G(V, W) \\ f \mapsto f - f(0) \end{array} \quad \text{and} \quad \tau_G : \begin{array}{c} \mathrm{Aff}_G(V, W) \to W^G \\ f \mapsto f(0). \end{array}$$

Theorem 6 implies that $\theta_G$ is an isomorphism and a similar proof to the one of Proposition 7 shows that both $\lambda_G$ and $\tau_G$ are linear, equivariant, and surjective. When it will be clear we are working in the equivariant setting, we will drop the subscript and just write $\theta$, $\lambda$, and $\tau$.

In the main text we will often use the following results.

**Proposition 9.** *Let $V_1$, $V_2$, and $V$ be $G$-representations. Then,*

$$\mathrm{Hom}_G(V_1 \oplus V_2, V) = \mathrm{Hom}_G(V_1, V) \oplus \mathrm{Hom}_G(V_2, V)$$

*and*

$$\mathrm{Hom}_G(V, V_1 \oplus V_2) = \mathrm{Hom}_G(V, V_1) \oplus \mathrm{Hom}_G(V, V_2).$$

**Proposition 10.** *Let $V$ and $W$ be $G$-representations, and let $G$ act trivially on $\mathbb{R}^n$ and $\mathbb{R}^m$. Then,*

$$\operatorname{Hom}_G(V \otimes \mathbb{R}^n, W \otimes \mathbb{R}^m) \cong \operatorname{Hom}_G(V, W) \otimes \operatorname{Hom}(\mathbb{R}^n, \mathbb{R}^m),$$

*and*

$$(V \otimes \mathbb{R}^n)^G \cong V^G \otimes \mathbb{R}^n.$$

*In other words, recalling that $\operatorname{Aff}_G(V, W) \cong \operatorname{Hom}_G(V, W) \oplus W^G$ and since $\operatorname{Hom}(\mathbb{R}^n, \mathbb{R}^m) \cong \mathbb{R}^{n \times m}$ is the set of $n \times m$ matrices over $\mathbb{R}$, understanding the structure of $\operatorname{Aff}_G(V \otimes \mathbb{R}^n, W \otimes \mathbb{R}^m)$ reduces to understanding the structure of $\operatorname{Aff}_G(V, W)$.*

### A.5 ON PERMUTATION REPRESENTATIONS

The followings are easy and known results from the theory of permutation representations.

We recall the definition of permutation representations as stated in Section 4.1.

**Definition 9.** *Let $X$ be a finite set and $G$ a finite group acting on it. A* permutation representation *of $G$ is a representation of $G$ on $\mathbb{R}^X$ such that $g(e_x) = e_{gx}$ for each $g \in G$ and $x \in X$.*

**Proposition 11.** *Let $X$ and $Y$ be two $G$-sets. We have the two following $G$-representations isomorphisms*

$$\mathbb{R}^{X \sqcup Y} \cong \mathbb{R}^X \oplus \mathbb{R}^Y \text{ and } \mathbb{R}^{X \times Y} \cong \mathbb{R}^X \otimes \mathbb{R}^Y,$$

*where $X \sqcup Y$ indicate the disjoint union of the sets $X$ and $Y$.*

**Example 6.** *Let $S = [n]$ and let $S_n$ act on $S$ in the standard way and note that $\mathbb{R}^S \cong \mathbb{R}^n$ as representations. From Proposition 11, we obtain that tensors of order $2$ are $\mathbb{R}^n \otimes \mathbb{R}^n = \mathbb{R}^{S \times S} = \mathbb{R}^{n \times n}$. Let $\Delta = \{(i, i) \mid i \in S\}$ and $\overline{\Delta} = \{(i, j) \in S \mid i \neq j\}$, note that $S \times S = \Delta \sqcup \overline{\Delta}$ and that $S_n$ acts transitively on both $\Delta$ and $\overline{\Delta}$. Therefore, $\mathbb{R}^n \otimes \mathbb{R}^n \cong \mathbb{R}^{S \times S} \cong \mathbb{R}^\Delta \oplus \mathbb{R}^{\overline{\Delta}}$.*

We say that a group $G$ acts transitively on a set $X$ if this action has only one orbit, namely $Gx = X$ for each $x \in X$. If $X = X_1 \sqcup \cdots \sqcup X_n$ is the orbit decomposition of $X$, then $\mathbb{R}^X \cong \mathbb{R}^{X_1} \oplus \cdots \oplus \mathbb{R}^{X_n}$.

As the bias terms of equivariant layers are vectors invariant under the action of permutation representations, it is important to characterize the invariant part of a permutation representation. Before proceeding, it is necessary to state the following result.

**Proposition 12.** *If $G$ is a finite group acting transitively on a finite set $X$, then there exists a subgroup $H < G$ and a $G$-set bijection between $X$ and $G/H$.*

Proposition 12 implies that we can restrict our study to representations of the form $\mathbb{R}^{G/H}$ for some subgroup $H$ of $G$. With this result, we can now proceed to prove Proposition 1. Let $G$ be a group and $X$ be a finite set with action of $G$. For $Y \subseteq X$, recall that $\mathbb{1}_Y = \sum_{y \in Y} e_y$.

*Proof of Proposition 1.* The Reynolds operator

$$\mathcal{R}: \quad \begin{aligned} V &\longrightarrow V^G \\ v &\mapsto \sum_{g \in G} gv \end{aligned}$$

projects each $G$-representation $V$ on its invariant subspace $V^G$. In the case $V = \mathbb{R}^{G/H}$, $e_{kH}$ is an element of the canonical base of $\mathbb{R}^{G/H}$,

$$\mathcal{R}(e_{kH}) = \sum_{g \in G} g e_{kH} = \sum_{g \in G} e_{gkH} = |H| \sum_{gH \in G/H} e_{gH} = |H| \mathbb{1}_{G/H}.$$

The final observation follows from Proposition 11. $\qquad\square$

Propositions 11 and 9 together imply that characterizing equivariant maps between permutation representations reduces to characterizing equivariant maps between representations induced by transitive actions on finite sets, or equivalently, left cosets by Proposition 12. We address this in Proposition 14. To prove this result, we first define the concept of right multiplication in Definition 10 and then prove Proposition 13, which characterizes equivariant maps between regular representations.

**Definition 10.** *For each $g \in G$ define the right-multiplication*

$$\mathcal{R}_g : \quad \begin{aligned} \mathbb{R}^G &\longrightarrow \mathbb{R}^G \\ e_h &\mapsto e_{hg^{-1}}. \end{aligned}$$

**Proposition 13.** *Right actions are a basis for the space of equivariant endomorphisms of the regular representation. In other words, $\{\mathcal{R}_g\}_{g \in G}$ is a basis for $\mathrm{Hom}_G(\mathbb{R}^G, \mathbb{R}^G)$.*

*Proof.* Each linear application $\phi \in \mathrm{Hom}(\mathbb{R}^G, \mathbb{R}^G)$ is defined by the values $\phi(e_g)$ for each $g \in G$ by linear extension. If $\phi$ is $G$-equivariant, it is defined just by its value on $e_e$. Indeed, $\phi(e_g) = \phi(ge_e) = g\phi(e_e)$ for each $g \in G$. Note that $\mathcal{R}_g$ is linear as the right action of $g$ on $\mathbb{R}^G$ is linear and $\mathcal{R}_g(e_h) = e_{hg^{-1}} = e_h g^{-1}$. It is also equivariant, indeed $\mathcal{R}_g(hv) = hvg^{-1} = h\,\mathcal{R}_g(v)$ for each $v \in \mathbb{R}^G$ and $h \in G$. Furthermore, $\mathcal{R}_{g^{-1}}(e_e) = e_g$ for each $g \in G$ and therefore they generate $\mathrm{Hom}_G(\mathbb{R}^G, \mathbb{R}^G)$.

Suppose that there exist values $a_g \in \mathbb{R}$ for each $g \in G$ such that $\sum_{g \in G} a_g \mathcal{R}_g = 0$, then

$$0 = \sum_{g \in G} a_g \mathcal{R}_g(e_e) = \sum_{g \in G} a_g e_{g^{-1}}.$$

Since elements $e_g$ are linearly independent, $a_g = 0$ for each $g \in G$. Hence, $\mathcal{R}_g$ are linearly independent and form a basis for $\mathrm{Hom}_G(\mathbb{R}^G, \mathbb{R}^G)$. $\qquad\square$

Now we would like to have a result similar to Proposition 13 but for morphisms between $G/K$ and $G/H$. To do this we need to define the following injection

$$\iota_{G/H} : \quad \begin{aligned} \mathbb{R}^{G/H} &\to \mathbb{R}^G \\ e_{gH} &\mapsto \frac{1}{|H|} \sum_{h \in H} e_{gh}, \end{aligned}$$

and projection

$$\pi_{G/H} : \quad \begin{aligned} \mathbb{R}^G &\to \mathbb{R}^{G/H} \\ e_g &\mapsto e_{gH}. \end{aligned}$$

For an arbitrary representation $V$, we define two surjective maps

$$\iota_{G/K}^* : \quad \begin{aligned} \mathrm{Hom}_G(\mathbb{R}^G, V) &\twoheadrightarrow \mathrm{Hom}_G(\mathbb{R}^{G/K}, V) \\ \phi &\mapsto \phi \circ \iota_{G/K}, \end{aligned}$$

and

$$\pi_{G/H*} : \quad \begin{aligned} \mathrm{Hom}_G(V, \mathbb{R}^G) &\twoheadrightarrow \mathrm{Hom}_G(V, \mathbb{R}^{G/H}) \\ \phi &\mapsto \pi_{G/H} \circ \phi. \end{aligned}$$

We can now generalize the concept of right multiplication to the general case of transitive actions, a concept necessary for stating Proposition 14, which we will then prove by following the approach used in the proof of Proposition 13.

**Definition 11.** *Define $\mathcal{R}_{HgK} = \alpha\,\mathcal{R}_g$ where the map $\alpha = \pi_{G/H*} \circ \iota_{G/K}^*$ is defined from $\mathrm{Hom}_G(\mathbb{R}^G, \mathbb{R}^G)$ to $\mathrm{Hom}_G(\mathbb{R}^{G/K}, \mathbb{R}^{G/H})$.*

**Proposition 14.** *The map $\mathcal{R}_{HgK}$ is well-defined and the set $\{\mathcal{R}_{HgK}\}_{HgK \in H \backslash G/K}$ is a basis for $\mathrm{Hom}_G(\mathbb{R}^{G/K}, \mathbb{R}^{G/H})$. Finally,*

$$\left( \mathcal{R}_{HgK}(e_{kK}) \right)_{sH} = \begin{cases} \frac{1}{|K|} & \text{if } sH \subseteq kKg^{-1}H, \\ 0 & \text{otherwise.} \end{cases}$$

*Proof.* To prove that $\mathcal{R}_{HgK}$ is well-defined, we need to prove that $\alpha \mathcal{R}_g = \alpha \mathcal{R}_{hgk}$ for each $h \in H$ and $k \in K$. Indeed,

$$\pi_{G/H} \mathcal{R}_{hgk} \iota_{G/K}(e_{sK}) = \frac{1}{|K|} \sum_{t \in K} e_{stk^{-1}g^{-1}h^{-1}H} =$$

$$\frac{1}{|K|} \sum_{t \in K} e_{stk^{-1}g^{-1}h^{-1}H} = \frac{1}{|K|} \sum_{t \in K} e_{stg^{-1}H} = \pi_{G/H} \mathcal{R}_g \iota_{G/K}(e_{sK}),$$

where the penultimate equality is true because $h^{-1}H = H$ and variable change $t \mapsto tk^{-1}$ in the sum.

By Proposition 13 the set $\{\mathcal{R}_g\}_{g \in G}$ is a basis for $\text{Hom}_G(\mathbb{R}^G, \mathbb{R}^G)$. By the previous observation we have shown that the image of $\{\mathcal{R}_g\}_{g \in G}$ under $\alpha$ is $\{\mathcal{R}_{HgK}\}_{HgK \in H \backslash G / K}$. As $\alpha$ is a surjection, $\{\mathcal{R}_{HgK}\}_{HgK \in H \backslash G / K}$ generates $\text{Hom}_G(\mathbb{R}^{G/K}, \mathbb{R}^{G/H})$.

Proving linear independence is similar to the proof of linear independence in Proposition 13. Indeed, let $a_{HgK} \in \mathbb{R}$ for each $HgK \in H \backslash G / K$ such that

$$\sum_{HgK \in H \backslash G / K} a_{HgK} \mathcal{R}_{HgK} = 0.$$

Hence,

$$0 = \sum_{HgK \in H \backslash G / K} a_{HgK} \mathcal{R}_{HgK}(e_K) = \sum_{HgK \in H \backslash G / K} \frac{1}{|K|} a_{HgK} \sum_{t \in K} e_{tg^{-1}H}.$$

Note that sets $\{tg^{-1}H\}_{t \in K}$ are pairwise disjoint with $g$ varying between representatives of $HgK$. This means that the respective vectors $\sum_{t \in K} e_{tg^{-1}H}$ are linearly independent, hence each $a_{HgK} = 0$. This proves that the maps $\mathcal{R}_{HgK}$ are linearly independent. Finally, observing that

$$\mathcal{R}_{HgK}(e_{kK}) = \frac{1}{|K|} \sum_{t \in K} e_{ktg^{-1}H},$$

it is clear that

$$\left( \mathcal{R}_{HgK}(e_{kK}) \right)_{sH} = \begin{cases} \frac{1}{|K|} & \text{if } sH \subseteq kKg^{-1}H, \\ 0 & \text{otherwise.} \end{cases}$$

$\square$

*Remark* 2. In our case of interest, in which $G$ is a finite group, the map $v \mapsto v \cdot w$ is equivalent at convolving $v$ by $w$. Proposition 14 is just a restatement and integration of Theorem 1 in Kondor & Trivedi (2018) in the restricted case of homogeneous spaces of finite groups.

# B MAIN RESULTS

## B.1 THE CHARACTERIZATION THEOREM

To formally state and prove Theorem 1, we need to understand how bias terms in neural spaces transform under the twin network trick. Note that, in general, we have $\tau(\overline{\text{Aff}_G(V, W)}) \subsetneq \tau(\text{Aff}_G(V \oplus V, W \oplus W))$, as illustrated by the following Example 7.

**Example 7** (Twin Layers in 2-IGNs). *Let us consider IGN layers. In this case, $V = W = \mathbb{R}^X$. The twin layer takes and has values in $\mathbb{R}^X \oplus \mathbb{R}^X \cong \mathbb{R}^{X \sqcup X'}$ where $X'$ is a disjoint copy of $X$. Hence, the twin IGN layer is an affine function $\mathbb{R}^{X \sqcup X'} \to \mathbb{R}^{X \sqcup X'}$ defined as follows:*

$$(A, B) \mapsto (L(A), L(B)) + y_1 \mathbb{1}_{X_1} + y_2 \mathbb{1}_{X_2} + y_1 \mathbb{1}_{X_1'} + y_2 \mathbb{1}_{X_2'}, \tag{5}$$

*where $X_1'$ and $X_2'$ are the copies of $X_1$ and $X_2$ in $X'$, respectively. Note that they share the same parameter for $\mathbb{1}_{X_1}$ and $\mathbb{1}_{X_1'}$ and for $\mathbb{1}_{X_2}$ and $\mathbb{1}_{X_2'}$, while the translational part of $\text{Aff}_{S_n}(\mathbb{R}^{X \sqcup X'}, \mathbb{R}^{X \sqcup X'})$ is $y_1 \mathbb{1}_{X_1} + y_2 \mathbb{1}_{X_2} + y_1' \mathbb{1}_{X_1'} + y_2' \mathbb{1}_{X_2'}$. Alternatively, we have*

$$\dim \tau(\overline{\text{Aff}_{S_n}(\mathbb{R}^X, \mathbb{R}^X)}) = 2 \text{ and } \dim \tau(\text{Aff}_{S_n}(\mathbb{R}^{X \sqcup X'}, \mathbb{R}^{X \sqcup X'})) = 4.$$

*Hence,*

$$\tau \left( \overline{\text{Aff}_{S_n}(\mathbb{R}^X, \mathbb{R}^X)} \right) \subsetneq \tau \left( \text{Aff}_{S_n}(\mathbb{R}^{X \sqcup X'}, \mathbb{R}^{X \sqcup X'}) \right).$$

Example 7 shows that we need a definition of bias that can describe the bias terms in (5). This will be provided in Definition 12.

**Definition 12** (Complete Bias). *We say that the subspace $M$ of $\mathrm{Aff}_G(V, \mathbb{R}^X)$ has* complete bias *if $\lambda(M) \oplus \tau(M) \cong M$ through $\theta$ and $\tau(M) = \langle \mathbb{1}_P \rangle_{P \in \mathcal{P}}$, where $\mathcal{P}$ is a partition of $X$. In this case, we say that the bias of $M$ is* subordinate *to the partition $\mathcal{P}$. In the opposite case, we say that $M$ has* incomplete bias. *In particular, we say that $M$ has* null bias *if its translational part $\tau(M)$ is zero; in other words, $M$ is simply a subspace of $\mathrm{Hom}_G(V, \mathbb{R}^X)$ and $M = \lambda(M)$.*

The following observations are necessary to justify this definition.

*Remark* 3. Note that each subspace $M$ of $\mathrm{Aff}_G(V, \mathbb{R}^X)$ such that $M = \lambda(M) \oplus \tau(M)$ and $\tau(M) = \tau(\mathrm{Aff}_G(V, \mathbb{R}^X))$ has complete bias. Indeed, in this case we have $\mathcal{P} = \{X_1, \ldots, X_n\}$, the $G$-orbit decomposition of $X$ and
$$\tau(M) = \langle \mathbb{1}_{X_i} \rangle_{X_i \in \mathcal{P}}$$
as proven in Proposition 1. Therefore, all examples in Section 4.2 have complete bias.

*Remark* 4. Note that each subspace $M$ of $\mathrm{Aff}_G(V, \mathbb{R}^X)$ has complete bias if and only if there exist $\phi_1, \ldots, \phi_s$ in $\mathrm{Hom}_G(V, \mathbb{R}^X)$ and a partition $\mathcal{P}$ of $X$ such that each map in $M$ can be written as
$$\beta \mapsto x_1 \phi_1(\beta) + \cdots + x_s \phi_s(\beta) + \sum_{Y \in \mathcal{P}} y_P \mathbb{1}_Y \tag{6}$$
where $x_i$ and $y_P$ are real parameters for each $i = 1, \ldots, s$ and each $P$ in $\mathcal{P}$.

**Proposition 15.** *Let $M_1, \ldots, M_{d-1}$ be subspaces with complete bias, then the space of twin networks*
$$\mathcal{N}_\sigma(\overline{M}_1, \ldots, \overline{M}_{d-1}, M'_d)$$
*has intermediate layers with complete bias and output layer with null bias.*

The proof of Proposition 15 relies on Proposition 16 and Lemma 1 which will be stated shortly.

**Definition 13.** *Let $X$ be a finite set, we define the* duplicate set *of $X$ as the set $X \sqcup X'$ where $X'$ is a disjoint copy of $X$. Let $\mathcal{P}$ be a partition of $X$, we define the* duplicate partition *of $\mathcal{P}$ as the partition $\mathcal{P}'$ of the duplicate of $X$ such that $\mathcal{P}' = \{Y \sqcup Y' \mid Y \in \mathcal{P}\}$. For each $y \in Y$, we will usually indicate the respective element in $Y'$ as $y'$, although when it will be clear from the context we may abuse notation and call both $y$.*

**Proposition 16.** *If $M$ is a sub-vector space of $\mathrm{Aff}_G(V, \mathbb{R}^X)$ with complete bias subordinate to partition $\mathcal{P}$ then the twin space $\overline{M}$ is a subspace of $\mathrm{Aff}_G(V \oplus V, \mathbb{R}^X \oplus \mathbb{R}^X) \cong \mathrm{Aff}_G(V \oplus V, \mathbb{R}^{X \sqcup X'})$ and has complete bias subordinate to the duplicate partition of $\mathcal{P}$. In particular, $\dim \tau(\overline{M}) = \dim \tau(M)$ and $\dim \tau(\mathrm{Aff}_G(V \oplus V, \mathbb{R}^X \oplus \mathbb{R}^X)) = 2 \dim \tau(\mathrm{Aff}_G(V, \mathbb{R}^X)) = 2 \dim \tau(\overline{\mathrm{Aff}_G(V, \mathbb{R}^X)})$.*

*Proof.* Trivially, if $M = \lambda(M) \oplus \tau(M)$, then $\overline{M} = \lambda(\overline{M}) \oplus \tau(\overline{M})$. Noticing that $\zeta : \mathbb{R}^X \oplus \mathbb{R}^X \cong \mathbb{R}^{X \sqcup X'}$ such that $\zeta((e_x, 0)) = e_x$ and $\zeta((0, e_x)) = e_{x'}$ is an isomorphism. First we show that that $\mathrm{Aff}_G(V \oplus V, \mathbb{R}^X \oplus \mathbb{R}^X) \cong \mathrm{Aff}_G(V \oplus V, \mathbb{R}^{X \sqcup X'})$. If $M$ has complete bias subordinate to $\mathcal{P}$ then $\tau(M) = \langle \mathbb{1}_Y \rangle_{Y \in \mathcal{P}}$ and each $v \in \tau(M)$ can be written as $v = \sum_{Y \in \mathcal{P}} a_Y \mathbb{1}_Y$ for some $(a_Y)_Y \in \mathbb{R}^{\mathcal{P}}$. Remember that $\tau(M) = \{\phi(0) \mid \phi \in M\}$, then
$$\tau(\overline{M}) = \{(\phi(0), \phi(0)) \mid \phi \in M\} = \{(v, v) \mid v \in \tau(M)\} =$$
$$\left\{ \left( \sum_{Y \in \mathcal{P}} a_Y \mathbb{1}_Y, \sum_{Y \in \mathcal{P}} a_Y \mathbb{1}_Y \right) \mid (a_Y)_Y \in \mathbb{R}^{\mathcal{P}} \right\}.$$
By the isomorphism $\zeta : \mathbb{R}^X \oplus \mathbb{R}^X \cong \mathbb{R}^{X \sqcup X'}$, we can write $\zeta(\sum_{Y \in \mathcal{P}} a_Y \mathbb{1}_Y, \sum_{Y \in \mathcal{P}} a_Y \mathbb{1}_Y) = \sum_{Y \in \mathcal{P}} a_Y \mathbb{1}_{Y \sqcup Y'}$. □

**Lemma 1.** *The output space of a twin network $M'$ always has null bias, independently of the bias space $\tau(M)$.*

*Proof.* Let $\phi : v \mapsto f(v) + y$ be an affine map in $M < \mathrm{Aff}_G(V, W)$. Then the output layer of the twin network is
$$\phi'(v, w) = \phi(v) - \phi(w) = f(v) + y - f(w) - y = f(v) - f(w),$$
which is always a linear map in $\mathrm{Aff}_G(V \oplus V, W)$. Equivalently, $\tau(M') = 0$. □

*Proof of Proposition 15.* Apply Proposition 16 to $M_1, \ldots, M_{d-1}$ and Lemma 1 to $M_d$. $\qquad \square$

Together Remark 3 and Proposition 15 show that we only need to solve zero locus problems for networks with complete bias in the intermediate layers and null bias in the final layer. We are now able to give the complete and formal statement and proof of Theorem 1.

**Theorem 7.** *Let $M_1, \ldots, M_{d-1}$ have complete bias and let $M_d$ have null bias. Let $\phi^{d,1}, \ldots, \phi^{d,s_d}$ be a set of generators of $M_d < \mathrm{Aff}_G(\mathbb{R}^{X_d}, \mathbb{R}^{X_{d+1}})$, and let the bias of $M_{d-1}$ be subordinate to the partition $\mathcal{P}$. Furthermore, for each $h = 1, \ldots, s_d$ and each $k \in X_{d+1}$ define*

$$\Psi_{h,k} = \left\{ \mathcal{Q} \le \mathcal{P} \mid \sum_{i \in P} \phi_{ki}^{d,h} = 0, \, \forall P \in \mathcal{Q} \right\}.$$

*If $\sigma$ is a non-polynomial activation function, then we have the following recursive formula with respect to network depth*

$$\mathcal{I}(\mathcal{N}_\sigma(M_1, \ldots, M_d)) = \bigcap_{h,k} \bigcup_{\mathcal{Q} \in \Psi_{h,k}} \bigcap_{\substack{P \in \mathcal{Q} \\ i,j \in P}} \mathcal{I}(\mathcal{N}_\sigma(M_1, \ldots, M_{d-2}, (M_{d-1})_{ij})),$$

*where $(M_{d-1})_{ij} = \{ \phi' : x \mapsto \pi_i \phi(x) - \pi_j \phi(x) \mid \phi \in \lambda(M_{d-1}) \}$, $\pi_i : \mathbb{R}^X \to \mathbb{R}$ is the projection on the $i$-th component of $\mathbb{R}^X$ for each $i$ in $X$, and $\lambda(M_{d-1})$ is the linear part of $M_{d-1}$.*

*Proof.* Denote $\mathcal{F}_d = \{ \phi^{d,1}, \ldots, \phi^{d,s_d} \}$. We can restrict to compute $\mathcal{I}(\mathcal{N}_\sigma(M_1, \ldots, M_{d-1}, \mathcal{F}_d))$ since, by Lemma 7, we know

$$\mathcal{I}(\mathcal{N}_\sigma(M_1, \ldots, M_d)) = \mathcal{I}(\mathcal{N}_\sigma(M_1, \ldots, M_{d-1}, \mathcal{F}_d)).$$

Each $d$-layer neural network $\eta^{d,h}$ in $\mathcal{N}_\sigma(M_1, \ldots, M_{d-1}, \mathcal{F}_d)$ can be written, for each input $\beta$, as

$$\eta^{d,h}(\beta) = \phi^{d,h} \tilde{\sigma}(\eta^{d-1}(\beta) + y) \qquad\qquad (\forall h = 1, \ldots, s_d) \qquad (7)$$

where

- The map $\phi^{d,h}$ is the $h$-th element in $\mathcal{F}_d$ and is linear since $M_d$ has null bias.

- The the map $\eta^{d-1}$ is $(d-1)$-layer network belonging to $\mathcal{N}_\sigma(M_1, \ldots, M_{d-2}, \lambda(M_{d-1}))$.

- The vector $y$ is a bias term in the translational part of $M_{d-1}$, namely the invariant sub-space of $\mathbb{R}^{X_d}$, and has complete bias subordinate to a partition $\mathcal{Q}$. Hence,

$$y = \sum_{P \in \mathcal{Q}} y_P \mathbb{1}_P. \qquad (8)$$

In a similar fashion, define $\eta^{d-1,t}$ in $\mathcal{N}(M_1, \ldots, M_{d-1}, \lambda(M_{d-2}))$ for each $t = 1, \ldots, s_{d-1}$ and some $s_{d-1} \ge 1$. Note that, due to Remark 4,

$$\eta^{d-1} = \sum_{t=1}^{s_{d-1}} x_t \eta^{d-1,t} \qquad (9)$$

for some $x_1, \ldots, x_{s_{d-1}} \in \mathbb{R}$. Therefore, by substituting both (8) and (9) into (7), we get

$$\eta^{d,h}(\beta) = \phi^{d,h} \tilde{\sigma} \left( \sum_{t=1}^{s_{d-1}} x_t \eta^{d-1,t}(\beta) + y \right). \qquad (10)$$

Recall that $\phi^{d,h}$ is a linear map from $\mathbb{R}^{X_d}$ to $\mathbb{R}^{X_{d+1}}$, defined by the elements $\phi_{ki}^{d,h} = \phi_k^{d,h}(e_i)$ for each input entry $i \in X_d$ and output entry $k \in X_{d+1}$.

With this notation, we can express (10) in coordinates as follows

$$\eta_k^{d,h}(\beta) = \sum_{i \in X_d} \phi_{ki}^{d,h} \sigma \left( \sum_{t=1}^{s_{d-1}} x_t \eta_i^{d-1,t}(\beta) + y_i \right). \qquad (11)$$

For each $i \in X_d$, let $P$ be the unique element in $\mathcal{Q}$ containing $i$. Then $y_i = y_P$, where $y_i$ is the coefficient defined in (11), and $y_P$ is the one defined in (8).

Hence, we can write (11) as follows

$$\eta_k^{d,h}(\beta) = \sum_{\substack{P \in \mathcal{Q} \\ i \in P}} \phi_{ki}^{d,h} \sigma \left( \sum_{t=1}^{s_{d-1}} x_t \eta_i^{d-1,t}(\beta) + y_P \right)$$

for each output entry $k$ in $\mathbb{R}^{X_{d+1}}$.

Thus, an element $\beta$ belongs to $\mathcal{I}(\mathcal{N}_\sigma(M_1, \ldots, M_{d-1}, \mathcal{F}_d))$ if and only if

$$\sum_{\substack{P \in \mathcal{Q} \\ i \in P}} \phi_{ki}^{d,h} \sigma \left( \sum_{t=1}^{s_{d-1}} x_t \eta_i^{d-1,t}(\beta) + y_P \right) = 0 \tag{12}$$

for each $x_t, y_P, h, k$, and $\eta^{d-1,t}$.

Assuming that $\sigma$ is non-polynomial and setting $a_i = \phi_{ki}^{d,h}$ and $b_i = (\eta_i^{d-1,t}(\beta))_t$, the second part of Theorem 9 implies that $(\eta_i^{d-1,t}(\beta))_{i,t}$ solves (12) for specific $h$ and $k$ if and only if

$$(\eta_i^{d-1,t}(\beta))_i \in \bigcup_{\mathcal{Q} \in \Psi_{h,k}} \bigcap_{\substack{P \in \mathcal{Q} \\ i,j \in P}} \left\{ (\eta_i^{d-1,t}(\gamma))_i \mid \eta_i^{d-1,t}(\gamma) - \eta_j^{d-1,t}(\gamma) = 0 \right\}.$$

Note that $\beta$ satisfies (12) for specific $h$ and $k$ if and only if $(\eta^{d-1,t}(\beta))_{i,t}$ satisfies it. Hence,

$$\beta \in \bigcup_{\mathcal{Q} \in \Psi_{h,k}} \bigcap_{\substack{P \in \mathcal{Q} \\ i,j \in P}} \left\{ \gamma \mid \eta_i^{d-1,t}(\gamma) - \eta_j^{d-1,t}(\gamma) = 0 \ \forall t \right\}. \tag{13}$$

By the definition of $(M_{d-1})_{ij}$, we get

$$\mathcal{I}(\mathcal{N}_\sigma(M_1, \ldots, M_{d-2}, (M_{d-1})_{ij})) = \{\beta \mid \eta_i^{d-1,t}(\beta) - \eta_j^{d-1,t}(\beta) = 0\}.$$

Therefore, $\beta$ satisfies (12) for specific $h$ and $k$ if and only if

$$\beta \in \bigcup_{\mathcal{Q} \in \Psi_{h,k}} \bigcap_{\substack{P \in \mathcal{Q} \\ i,j \in P}} \mathcal{I}(\mathcal{N}_\sigma(M_1, \ldots, M_{d-2}, (M_{d-1})_{ij})).$$

Since $\beta$ has to satisfy (12) for each $h$ and $k$, we finally get

$$\mathcal{I}(\mathcal{N}_\sigma(M_1, \ldots, M_d)) = \bigcap_{h,k} \bigcup_{\mathcal{Q} \in \Psi_{h,k}} \bigcap_{\substack{P \in \mathcal{Q} \\ i,j \in P}} \mathcal{I}(\mathcal{N}_\sigma(M_1, \ldots, M_{d-2}, (M_{d-1})_{ij})).$$

$\square$

*Remark* 5. Theorem 1 could actually be stated with different activation functions for each layer, as long as they are all non-polynomial. However, for readability and simplicity, we have presented the results using a single activation function.

*Remark* 6. Here, we demonstrate that the complete bias assumption is necessary for all non-polynomial activations to achieve maximal separation power. Specifically, let us examine the separation power of the set of shallow neural networks where all representation spaces are one-dimensional and the hidden layer has a null, and therefore incomplete, bias term. The main concern is the separability of opposite inputs $\beta$ and $-\beta$. This reduces to study the identification equation

$$y\sigma(\beta x) = y\sigma(-\beta x)$$

for each $x, y \in \mathbb{R}$. Any even function $\sigma$, including non-polynomial ones, solves this equation but does not achieve maximal separation power, which could be reached by adding a bias term, as shown in Theorem 2.

## B.2 THE ROLE OF ACTIVATIONS

*Proof of Theorem 2.* We prove that non-polynomial activation functions have equivalent separation power by induction on $d$.

If $d = 1$ then $\mathcal{I}(\mathcal{N}_\sigma(M_1)) = \mathcal{I}(M_1)$ which does not depend on $\sigma$.

Now suppose that $\mathcal{I}(\mathcal{N}_\sigma(M_1, \ldots, M_{d-1}))$ does not depend on $\sigma$ for each sequence $M_1, \ldots, M_{d-1}$. Then, observing (3)

$$\mathcal{I}(\mathcal{N}_\sigma(M_1, \ldots, M_d)) = \bigcap_{h,k} \bigcup_{\mathcal{P} \in \Psi_{h,k}} \bigcap_{\substack{P \in \mathcal{P} \\ i,j \in P}} \mathcal{I}(\mathcal{N}_\sigma(M_1, \ldots, M_{d-2}, (M_{d-1})_{ij})),$$

we note that $\mathcal{I}(\mathcal{N}_\sigma(M_1, \ldots, M_d))$ is independent of $\sigma$ as indices such as $h, k$ and $i, j$ are independent of $\sigma$, as well as $\mathcal{I}(\mathcal{N}_\sigma(M_1, \ldots, M_{d-2}, (M_{d-1})_{ij}))$ is by inductive hypothesis.

Finally, the first part of Theorem 2 follows directly from the proof of Theorem 1 and the last part of Theorem 9. $\square$

## B.3 THE ROLE OF DEPTH

*Proof of Theorem 3.* To prove the first part of the statement, by Lemma 5, it suffices to show that

$$\mathcal{N}_\sigma(M_1, \ldots, M_i, \ldots, M_d) \subseteq \mathcal{N}_\sigma(M_1, \ldots, \underbrace{M_i, \ldots, M_i}_{n\text{-times}}, \ldots, M_d). \tag{14}$$

for each $n \geq 1$.

Moreover, Theorem 2 implies that is enough to prove this inclusion for a fixed non-polynomial $\sigma$; then Theorem 3 will hold for any other non-polynomial activation as well. Therefore, let $\sigma$ be the ReLU activation function, noting that in this case $\sigma \circ \sigma = \sigma$.

In particular,

$$\tilde{\sigma} = \underbrace{\tilde{\sigma} \circ \tilde{\sigma} \circ \cdots \circ \tilde{\sigma}}_{n\text{-times}} = \underbrace{\tilde{\sigma} \circ id_{\mathbb{R}^{X_i}} \circ \tilde{\sigma} \circ \cdots \circ \tilde{\sigma} \circ id_{\mathbb{R}^{X_i}}}_{n\text{-times}},$$

for each $n \geq 1$.

Thus, each neural network $\phi_d \circ \tilde{\sigma} \circ \cdots \circ \tilde{\sigma} \circ \phi_1$ in $\mathcal{N}_\sigma(M_1, \ldots, M_i, \ldots, M_d)$ can be written as

$$\phi_d \circ \tilde{\sigma} \circ \cdots \circ \tilde{\sigma} \circ \phi_{i+1} \circ \underbrace{\tilde{\sigma} \circ id_{\mathbb{R}^{X_i}} \circ \tilde{\sigma} \circ \cdots \circ \tilde{\sigma} \circ id_{\mathbb{R}^{X_i}}}_{n\text{-times}} \circ \tilde{\sigma} \circ \phi_{i-1} \circ \cdots \circ \tilde{\sigma} \circ \phi_1$$

which is an element of $\mathcal{N}_\sigma(M_1, \ldots, \underbrace{M_i, \ldots, M_i}_{n\text{-times}}, \ldots, M_d)$, thereby proving (14).

The final step is to prove the stabilization property. This is achieved by recalling that, by Proposition 2 and Theorem 1,

$$\mathcal{I}(\mathcal{N}_\sigma(\overline{M}_1, \ldots, \overline{M}_{d-1}, M_d')) = \bigcap_{h,k} \bigcup_{\mathcal{P} \in \Psi_{h,k}} \bigcap_{\substack{P \in \mathcal{P} \\ i,j \in P}} \mathcal{I}(\mathcal{N}_\sigma(\overline{M}_1, \ldots, \overline{M}_{d-2}, (M_{d-1})_{ij})). \tag{15}$$

Define

$$C_n = \mathcal{I}(\mathcal{N}_\sigma(\overline{M}_1, \ldots, \underbrace{\overline{M}_i, \ldots, \overline{M}_i}_{n \text{ times}}, \ldots, M_d'))$$

for each $n \in \mathbb{N}$. Recursively applying (15), both $C_n$ and $C_m$ can be represented as unions and intersections of elements in the finite set

$$\mathcal{C} = \{\mathcal{I}(\mathcal{N}_\sigma(\overline{M}_1, \ldots, \overline{M}_{i-1}, (\overline{M}_i)_{hk}))\}_{h,k \in X_i}.$$

We can reformulate the descending sequence (14) as follows

$$\cdots \subseteq C_n \subseteq C_{n-1} \subseteq \cdots \subseteq C_1$$

which stabilizes due to Lemma 2. $\square$

**Lemma 2.** *Let $\mathcal{C} = \{C_1, \ldots, C_d\}$ be a finite collection of sets. The following statements are true:*

- *Let $\mathcal{C}_\cup = \{C_{i_1} \cup \cdots \cup C_{i_r} \mid 1 \leq i_1, \ldots, i_r \leq d, \ r \in \mathbb{N}\}$ be the collection of unions of a finite number of sets in $\mathcal{C}$. Then $\mathcal{C}_\cup$ is finite.*

- *Let $\mathcal{C}_\cap = \{C_{i_1} \cap \cdots \cap C_{i_r} \mid 1 \leq i_1, \ldots, i_r \leq d, \ r \in \mathbb{N}\}$ be the collection of intersections of a finite number of sets in $\mathcal{C}$. Then $\mathcal{C}_\cap$ is finite.*

- *Let $\tilde{\mathcal{C}}$ the smaller collection containing $\mathcal{C}$ which is closed by intersection and union. Then $\tilde{\mathcal{C}} = (\mathcal{C}_\cap)_\cup = (\mathcal{C}_\cup)_\cap$ and, in particular, is finite.*

*In particular, ascending and descending sequences of inclusions in $\tilde{\mathcal{C}}$ stabilize.*

*Proof.* To prove the first point, it is sufficient to note that duplicates in the expression $C_{i_1} \cup \cdots \cup C_{i_r}$ can be removed. Therefore, the cardinality of $\mathcal{C}_\cup$ is bounded by the number of possible tuples $i_1, \ldots, i_r$ which are $2^d$. The proof of the second point is analogous.

By the distributive property of intersections with respect to unions we obtain that each element in $\tilde{\mathcal{C}}$ can be written as
$$(C_{i_{1,1}} \cap \cdots \cap C_{i_{1,d_1}}) \cup \cdots \cup (C_{i_{r,1}} \cap \cdots \cap C_{i_{r,d_r}}).$$
Hence, $\tilde{\mathcal{C}} = (\mathcal{C}_\cap)_\cup$. Similarly, using the distributive property of unions with respect to intersections, we get $\tilde{\mathcal{C}} = (\mathcal{C}_\cup)_\cap$. In particular, $\tilde{\mathcal{C}}$ is finite as $\mathcal{C}_\cup$ and, hence, $(\mathcal{C}_\cup)_\cap$ are finite. $\qquad\square$

The repetition threshold may vary depending on the model and representation. For example, $k$-IGNs, being equivalent to $k$-WL, have a repetition threshold proportional to that of $k$-WL itself (Maron et al., 2019a; Geerts, 2020). In contrast, the Proposition 3 demonstrates an example of stabilization after just one repetition.

*Proof of Proposition 3.* From previous observations, we know that
$$\rho(\mathcal{N}_\sigma(V, \mathbb{R}^G, \ldots, \mathbb{R}^G, \mathbb{R}^{G/H})) \subseteq \rho(\mathcal{N}_\sigma(V, \mathbb{R}^G, \mathbb{R}^{G/H})). \tag{16}$$
Note that the family of equivariant continuous functions $\mathcal{C}_G(V, \mathbb{R}^{G/H})$ cannot separate $H$-orbits in $V$. Indeed, for each $f \in \mathcal{C}_G(V, \mathbb{R}^{G/H})$, $f(hv) = hf(v) = f(v)$ for each $h \in H$. Hence, Proposition 4 implies that $\mathcal{N}_\sigma(V, \mathbb{R}^G, \mathbb{R}^{G/H})$ has the finer separation power between families of functions in $\mathcal{C}_G(V, \mathbb{R}^{G/H})$. This implies equality in (16), concluding the proof. $\qquad\square$

### B.4    THE ROLE OF INTERMEDIATE REPRESENTATIONS

For now, we focus on developing the notation necessary to state and prove Theorem 4. The structure of our network of interest is as follows:
$$\eta : V \xrightarrow{\phi_1} V_1 \xrightarrow{\tilde{\sigma}} \cdots \xrightarrow{\phi_i} V_i' \oplus V_i'' \xrightarrow{\tilde{\sigma}} V_i' \oplus V_i'' \xrightarrow{\phi_{i+1}} \cdots \xrightarrow{\tilde{\sigma}} V_d \xrightarrow{\phi_{d+1}} W$$
with $\eta \in \mathcal{N}_\sigma(M_1, \ldots, M_d)$.

To formulate the identification equivalence relation of these networks in terms of the identification relations of simpler architectures with only $V'$ and $V''$ as intermediate representations, we need to define the projection map $\pi' : V' \oplus V'' \to V'$ and the immersion map $\iota' : V' \to V' \oplus V''$. Similarly, we can define $\pi''$ and $\iota''$. Furthermore, for any $G$-representation $W$, we define
$$\pi_*' : \begin{array}{c} \mathrm{Aff}_G(W, V' \oplus V'') \to \mathrm{Aff}_G(W, V') \\ f \mapsto \pi' \circ f \end{array} \quad \text{and} \quad \iota'^* : \begin{array}{c} \mathrm{Aff}_G(V' \oplus V'', W) \to \mathrm{Aff}_G(V', W) \\ f \mapsto f \circ \iota' \end{array}.$$
Similarly, we define $\pi_*''$ and $\iota''^*$. Let $M$ be a subspace of $\mathrm{Aff}_G(W, V' \oplus V'')$, its image $\pi_*'(M)$ is a subspace of $\mathrm{Aff}_G(W, V')$ and
$$M = \pi_*'(M) + \pi_*''(M).$$
Indeed, each $f \in M$ can be expressed as $f = \pi' f + \pi'' f = \pi_*'(f) + \pi_*''(f) f$, identifying $V'$ and $V''$ as subspaces of $V$. Similarly, for $M$ subspace of $\mathrm{Aff}_G(V' \oplus V'', W)$,
$$M = \iota'^*(M) + \iota''^*(M)$$

Hence, we can write

$$\mathcal{N}_\sigma(M_1, \ldots, M_d) = \mathcal{N}_\sigma(M_1, \ldots, \pi'_*(M_i) + \pi''_*(M_i), \iota'^*(M_{i+1}) + \iota''^*(M_{i+1}), \ldots, M_d),$$

and the problem informally stated above reduces to determining the separation power of the entire family $\mathcal{N}_\sigma(M_1, \ldots, M_d)$ by understanding the separation power of the smaller families $\mathcal{N}_\sigma(M_1, \ldots, \pi'M_i, \iota'^*(M_{i+1}), \ldots, M_d)$ and $\mathcal{N}_\sigma(M_1, \ldots, \pi''M_i, \iota''^*(M_{i+1}), \ldots, M_d)$. This is achieved by the following theorem.

**Theorem 8.** *With the notation defined above, we have*

$$\rho(\mathcal{N}_\sigma(M_1, \ldots, M_d)) =$$
$$\rho(\mathcal{N}_\sigma(M_1, \ldots, \pi'_*(M_i), \iota'^*(M_{i+1}), \ldots, M_d)) \cap \rho(\mathcal{N}_\sigma(M_1, \ldots, \pi''_*(M_i), \iota''^*(M_{i+1}), \ldots, M_d)).$$

*Proof.* Note that $\overline{\psi \circ \phi} = \overline{\psi} \circ \overline{\phi}$. Indeed, $\overline{\psi} \circ \overline{\phi} = (\phi, \phi) \circ (\psi, \phi) = (\phi \circ \psi, \phi \circ \psi) = \overline{\phi \circ \psi}$, and $\tilde{\sigma}\iota' = \iota'\tilde{\sigma}$ and $\tilde{\sigma}\pi' = \pi'\tilde{\sigma}$. Similarly, for $\pi''$ and $\iota''$.
Furthermore, $\overline{(\iota' + \iota'') \circ (\pi' + \pi'')} = (\overline{\iota'} + \overline{\iota''}) \circ (\overline{\pi'} + \overline{\pi''})$ since

$$\overline{(\iota' + \iota'') \circ (\pi' + \pi'')} = \overline{id_{V'_i \oplus V''_i}} = \left(\overline{id_{V'_i} \oplus 0_{V''_i}}\right) + \left(\overline{0_{V'_i} \oplus id_{V''_i}}\right)$$
$$= \left(\overline{\iota' \circ \pi'}\right) + \left(\overline{\iota'' \circ \pi''}\right) = \left(\overline{\iota'} \circ \overline{\pi'}\right) + \left(\overline{\iota''} \circ \overline{\pi''}\right)$$
$$= \left(\overline{\iota'} + \overline{\iota''}\right) \circ \left(\overline{\pi'} + \overline{\pi''}\right).$$

We now need to prove that

$$\overline{(\psi\iota' + \psi\iota'')} \, \tilde{\sigma} \, \overline{(\pi'\phi + \pi''\phi)} = \left(\overline{\psi\iota'} + \overline{\psi\iota''}\right) \tilde{\sigma} \left(\overline{\pi'\phi} + \overline{\pi''\phi}\right). \tag{17}$$

Indeed,

$$\overline{(\psi\iota' + \psi\iota'')} \, \tilde{\sigma} \, \overline{(\pi'\phi + \pi''\phi)} = \overline{\psi} \circ \overline{(\iota' + \iota'')} \tilde{\sigma} \overline{(\pi' + \pi'')} \circ \overline{\phi}$$
$$= \overline{\psi} \circ \overline{(\iota' + \iota'')(\pi' + \pi'')} \circ \tilde{\sigma}\overline{\phi} = \overline{\psi} \circ (\overline{\iota'} + \overline{\iota''})(\overline{\pi'} + \overline{\pi''}) \circ \tilde{\sigma}\overline{\phi}$$
$$= \overline{\psi} \circ (\overline{\iota'} + \overline{\iota''}) \tilde{\sigma} (\overline{\pi'} + \overline{\pi''})\overline{\phi} = \left(\overline{\psi\iota'} + \overline{\psi\iota''}\right) \tilde{\sigma} \left(\overline{\pi'\phi} + \overline{\pi''\phi}\right).$$

Hence, thanks to (17),

$$\mathcal{N}_\sigma(\overline{M}_1, \ldots, \overline{M}_{d-1}, M'_d) =$$
$$\mathcal{N}_\sigma(\overline{M}_1, \ldots, \overline{\pi'_*(M_i) + \pi''_*(M_i)}, \overline{\iota'^*(M_{i+1}) + \iota''^*(M_{i+1})}, \ldots, \overline{M}_{d-1}, M'_d) =$$
$$\mathcal{N}_\sigma(\overline{M}_1, \ldots, \overline{\pi'_*(M_i)} + \overline{\pi''_*(M_i)}, \overline{\iota'^*(M_{i+1})} + \overline{\iota''^*(M_{i+1})}, \ldots, \overline{M}_{d-1}, M'_d).$$

By Theorem 1 and the previous observations, we can limit to study spaces of the type

$$\mathcal{N}_\sigma(\overline{M}_1, \ldots, \overline{\pi'_*(M_i)} + \overline{\pi''_*(M_i)}, (\overline{\iota'^*(M_{i+1})} + \overline{\iota''^*(M_{i+1})})_{uv}) =$$
$$\mathcal{N}_\sigma(\overline{M}_1, \ldots, \overline{\pi'_*(M_i)} + \overline{\pi''_*(M_i)}, \overline{(\iota'^*(M_{i+1}))}_{uv}) +$$
$$\mathcal{N}_\sigma(\overline{M}_1, \ldots, \overline{\pi'_*(M_i)} + \overline{\pi''_*(M_i)}, \overline{(\iota''^*(M_{i+1}))}_{uv})$$

thanks to the linearity of the map $\phi \mapsto (\phi)_{uv}$. Note that

$$\mathcal{N}_\sigma(\overline{M}_1, \ldots, \overline{\pi'_*(M_i)} + \overline{\pi''_*(M_i)}, \overline{(\iota'^*(M_{i+1}))}_{uv}) = \mathcal{N}_\sigma(\overline{M}_1, \ldots, \overline{\pi'_*(M_i)}, \overline{(\iota'^*(M_{i+1}))}_{uv})$$

as $\pi' \circ \iota'' = 0$ and both projections and immersions commute with activations. From Lemma 6 we get

$$\mathcal{I}(\mathcal{N}_\sigma(\overline{M}_1, \ldots, \overline{\pi'_*(M_i)} + \overline{\pi''_*(M_i)}, (\overline{\iota'^*(M_{i+1})} + \overline{\iota''^*(M_{i+1})})_{uv})) =$$
$$= \mathcal{I}(\mathcal{N}_\sigma(\overline{M}_1, \ldots, \overline{\pi'_*(M_i)}, \overline{(\iota'^*(M_{i+1}))}_{uv}) \cap \mathcal{I}(\mathcal{N}_\sigma(\overline{M}_1, \ldots, \overline{\pi''_*(M_i)}, \overline{(\iota''^*(M_{i+1}))}_{uv}).$$

Combining all the above results, we conclude the proof of the theorem. $\square$

*Proof of Theorem 4.* Theorem 4 is a consequence of Theorem 8 in the case where $M_i$ is the full set $\text{Aff}_G(V_{i-1}, V_i)$. Note that

$$\pi'_* \text{Aff}_G(V_i, V'_i \oplus V''_i) = \text{Aff}_G(V_{i-1}, V'_i)$$

and

$$\iota'^* \text{Aff}_G(V'_i \oplus V''_i, V_{i+1}) = \text{Aff}_G(V'_i, V_{i+1}).$$

Similarly, for $\pi''_*$ and $\iota''^*$. $\square$

## B.5 THE ROLE OF REPRESENTATION TYPE

*Proof of Theorem 5.* Write $H/K = \{h_1 K, \ldots, h_s K\}$, we have the following injection

$$
\iota : \begin{array}{c} \mathbb{R}^{G/H} \longrightarrow \mathbb{R}^{G/K} \\ e_{gH} \mapsto \dfrac{1}{s} \sum_{i=1}^{s} e_{gh_i K} \end{array} \tag{18}
$$

and projection

$$
\pi : \begin{array}{c} \mathbb{R}^{G/K} \longrightarrow \mathbb{R}^{G/H} \\ e_{gK} \mapsto e_{gH}. \end{array} \tag{19}
$$

Note that $\pi\iota = id_{\mathbb{R}^{G/H}}$, indeed,

$$
\pi\iota(e_{gH}) = \frac{1}{s} \sum_{i=1}^{s} \pi(e_{gh_i K}) = \frac{1}{s} \sum_{i=1}^{s} e_{gh_i H} = e_{gH},
$$

as $gh_i H = gH$ for each $i = 1, \ldots, s$.

Consider the following diagram

$$
\eta : V \longrightarrow \cdots \xrightarrow{\phi} \mathbb{R}^{G/H} \xrightarrow{\sigma_H} \mathbb{R}^{G/H} \xrightarrow{\psi} \cdots \longrightarrow W
$$
$$
\pi \big\Updownarrow \iota \qquad \qquad \pi \big\Updownarrow \iota
$$
$$
\eta' : V \longrightarrow \cdots \xrightarrow{\phi'} \mathbb{R}^{G/K} \xrightarrow{\sigma'_H} \mathbb{R}^{G/K} \xrightarrow{\psi'} \cdots \longrightarrow W.
$$

From the network $\eta$ in $\mathcal{N}_\sigma(V, \ldots, \mathbb{R}^{G/H}, \ldots, W)$ composed by $\phi$, $\psi$, and $\sigma$ we want construct a new representation $\eta'$ defined as follows. Let $\phi' = \iota \circ \phi$, $\psi' = \psi \circ \pi$, and $\tilde{\sigma}' = \iota \circ \tilde{\sigma} \circ \pi$ and note that $\psi' \circ \tilde{\sigma}' \circ \phi' = \psi \circ \pi \circ \iota \circ \tilde{\sigma} \circ \pi \circ \iota \circ \phi = \psi \circ \tilde{\sigma} \circ \phi$. Hence, substituting $\psi \circ \tilde{\sigma} \circ \phi$ with $\psi' \circ \tilde{\sigma}' \circ \phi'$ inside the definition of $\eta$ do not change the function, and embeds it into a parameter space with intermediate representation $\mathbb{R}^{G/K}$ instead of $\mathbb{R}^{G/H}$. But to prove that $\eta$ is a neural network, we need to prove that $\tilde{\sigma}'$ is a point-wise activation function for some real-valued function $\sigma'$.

If $\tilde{\sigma}$ is a point-wise activation associated to $\sigma : \mathbb{R} \to \mathbb{R}$ defined on $\mathbb{R}^{G/H}$ we have that

$$
\tilde{\sigma}\Big( \sum_{gH \in G/H} a_{gH} e_{gH} \Big) = \sum_{gH \in G/H} \sigma(a_{gH}) e_{gH}.
$$

On the other hand, we have

$$
\tilde{\sigma}'\Big( \sum_{gK \in G/K} a_{gK} e_{gK} \Big) = \iota \circ \tilde{\sigma} \circ \pi \Big( \sum_{gK \in G/K} a_{gK} e_{gK} \Big) =
$$

$$
\iota \circ \tilde{\sigma}\Big( \sum_{\substack{gH \in G/H \\ ghK \in gH/K}} a_{ghK} e_{gH} \Big) = \iota \sum_{gH \in G/H} \sigma\Big( \sum_{ghK \in gH/K} a_{ghK} \Big) e_{gH} =
$$

$$
\frac{1}{s} \sum_{gH \in G/H} \sigma\Big( \sum_{ghK \in gH/K} a_{ghK} \Big) \sum_{hK \in H/K} e_{ghK} =
$$

$$
\frac{1}{s} \sum_{gK \in G/K} \sigma\Big( \sum_{hK \in H/K} a_{ghK} \Big) e_{gK}.
$$

Note that the map

$$
\alpha : \sum_{gK \in G/K} a_{gK} e_{gK} \mapsto \sum_{hK \in H/K} a_{ghK} e_{gK}
$$

is linear and $G$-equivariant. In particular, note that $\tilde{\sigma}' = \dfrac{\tilde{\sigma}_K \circ \alpha}{s}$, where we denote the standard point-wise activation induced by $\sigma$ on $\mathbb{R}^{G/K}$ as $\tilde{\sigma}_K$, to distinguish it from $\tilde{\sigma}$, the

point-wise activation induced by $\sigma$ but defined on $\mathbb{R}^{G/H}$. Hence, substituting $\psi \circ \tilde{\sigma} \circ \phi$ with $\psi' \circ \tilde{\sigma}' \circ \phi' = \psi' \circ \frac{\tilde{\sigma}_K \circ \alpha}{s} \circ \phi'$, we obtain an immersion of $\eta$ in $\mathcal{N}_\sigma(V, \ldots, \mathbb{R}^{G/K}, \ldots, W)$. Hence $\mathcal{N}_\sigma(V, \ldots, \mathbb{R}^{G/H}, \ldots, W) \subseteq \mathcal{N}_\sigma(V, \ldots, \mathbb{R}^{G/K}, \ldots, W)$ and $\rho(\mathcal{N}_\sigma(V, \ldots, \mathbb{R}^{G/K}, \ldots, W)) \subseteq \rho(\mathcal{N}_\sigma(V, \ldots, \mathbb{R}^{G/H}, \ldots, W))$ $\qquad\square$

We are now going to develop the tools to prove Proposition 4.

**Lemma 3.** *Let* $M = \mathrm{Aff}_G(V, \mathbb{R}^G)$*, then* $\mathcal{I}(M_{u,v}) = \mathcal{I}(M_{v^{-1}u,e}) = \mathcal{I}(M_{uv^{-1},e})$*. Moreover,* $(\beta, \beta') \in \mathcal{I}(M_{g,e})$ *if and only if* $g\beta = \beta'$*.*

*Proof.* Let $V = V_1 \oplus \cdots \oplus V_s$ where $V_i = \mathbb{R}^{G/K_i}$ for each $i = 1, \ldots, s$. By Proposition 14 and setting $H = \{e\}$, we know that $\mathrm{Hom}_G(V, \mathbb{R}^G) = \mathrm{Hom}_G(\mathbb{R}^{G/K_1}, \mathbb{R}^G) \oplus \cdots \oplus (\mathbb{R}^{G/K_s}, \mathbb{R}^G)$ is generated by functions $\mathcal{R}_{g_i K_i} \pi_{G/K_i}$ for each $g_i K_i \in G/K_i$ for each $i = 1, \ldots, s$, and $\pi_{G/K_i}$ is the projection of $V$ onto $V_i = \mathbb{R}^{G/K_i}$. Moreover,

$$\left( \mathcal{R}_{g_i K_i} \pi_{G/K_i}(\beta) \right)_u = \left( \mathcal{R}_{g_i K_i} \pi_{G/K_i} \left( \sum_{kK_i \in G/K_i} \beta_{kK_i} e_{kK_i} \right) \right)_u = \frac{1}{|K_i|} \beta_{ug_i K_i}.$$

For each $g \in G$, we have that

$$\mathcal{I}(M_{u,v}) =$$
$$\left\{ (\beta, \beta') \mid \left( \mathcal{R}_{g_i K_i} \pi_{G/K_i}(\beta) \right)_u - \left( \mathcal{R}_{g_i K_i} \pi_{G/K_i}(\beta') \right)_v = 0 \ \forall i \forall g_i K_i \in G/K_i \right\} =$$
$$\left\{ (\beta, \beta') \mid \beta_{ug_i K_i} - \beta'_{vg_i K_i} = 0 \ \forall i \forall g_i K_i \in G/K_i \right\} =$$
$$\left\{ (\beta, \beta') \mid v^{-1} u\beta = \beta' \right\}.$$

In particular, we have that $\mathcal{I}(M_{u,v}) = \mathcal{I}(M_{v^{-1}u,e})$. Hence, $(\beta, \beta') \in \mathcal{I}(M_{g,e})$ if and only if $g\beta = \beta'$.

Finally, in a similar way, we are able to observe that $\mathcal{I}(M_{u,v}) = \mathcal{I}(M_{uv^{-1},e})$.

$\qquad\square$

*Proof of Proposition 4.* In what follows we have to consider $G \sqcup G$, to distinguish the two distinct copies of $G$, we denote $G'$ as the second copy of $G$ and, and when $g$ is an element of $G$, we will indicate as $g'$ the analogous element in $G'$.

Define $M = \overline{\mathrm{Aff}_G(V, \mathbb{R}^G)}$ and $N = \mathrm{Aff}_G(\mathbb{R}^G, \mathbb{R}^{G/H})' < \mathrm{Hom}_G(\mathbb{R}^G \oplus \mathbb{R}^{G'}, \mathbb{R}^{G/H})$. Proposition 2 implies

$$\rho(\mathcal{N}_\sigma(V, \mathbb{R}^G, \mathbb{R})) = \mathcal{I}(\mathcal{N}_\sigma(M, N)).$$

Note that $N = \langle \mathcal{R}_{Hg} - \mathcal{R}_{H'g} \rangle_{Hg \in H \backslash G}$ where functions $\mathcal{R}_{Hg}$ are defined as

$$\left( \mathcal{R}_{Hg}(e_k) \right)_{sH} = \begin{cases} 1 & \text{if } s \in kg^{-1}H, \\ 0 & \text{otherwise.} \end{cases}$$

An element $\mathcal{Q}$ in $\Psi_{Hg,sH}$ is a partition of $G \sqcup G'$, where for each $P \in \mathcal{Q}$ the intersection $P \cap sHg \sqcup sH'g$ have the same number of elements in $sHg$ and $sH'g$. Due to Remark 8, we can just consider $\Psi'_{Hg,sH}$ containing the partitions of $G \sqcup G'$ whose only parts are $P = \{u, v\}$ for $u \in sHg$ and $v \in sH'g$, otherwise $P$ is a singleton not containing elements in $sHg$ or $sH'g$.

Hence, by Theorem 1,

$$\mathcal{I}(\mathcal{N}_\sigma(M, N)) = \bigcap_{Hg,sH} \bigcup_{\mathcal{Q} \in \Psi'_{Hg,sH}} \bigcap_{\{u,v\} \in \mathcal{Q}} \mathcal{I}(M_{u,v}). \tag{20}$$

If we prove that for each $Hg$ and $sH$

$$\bigcup_{\mathcal{Q} \in \Psi'_{Hg,sH}} \bigcap_{\{u,v\} \in \mathcal{Q}} \mathcal{I}(M_{u,v}) = \bigcup_{h \in H} \mathcal{I}(M_{h,e}). \tag{21}$$

then we are done. Indeed, thanks to Lemma 3, $(\beta, \beta') \in \bigcup_{h \in H} \mathcal{I}(M_{h,e})$ if and only if there exists some $h \in H$ such that $h\beta = \beta'$. Moreover, (21) does not depend on $Hg$ and $sH$ then the outer intersection in (20) is trivial.

Now to prove (21), we first show that

$$\bigcup_{\mathcal{Q} \in \Psi'_{Hg,sH}} \bigcap_{\{u,v\} \in \mathcal{Q}} \mathcal{I}(M_{u,v}) \subseteq \bigcup_{h \in H} \mathcal{I}(M_{h,e}).$$

Note that if $\{u, v\} \in \mathcal{Q}$ and $u, v \in sHg$, then, by Lemma 3,

$$\mathcal{I}(M_{u,v}) = \mathcal{I}(M_{shg,sh'g}) = \mathcal{I}(M_{hh'^{-1},e}).$$

Therefore,

$$\bigcap_{\{u,v\} \in \mathcal{Q}} \mathcal{I}(M_{u,v}) \subseteq \mathcal{I}(M_{u,v}) \subseteq \bigcup_{h \in H} \mathcal{I}(M_{h,e}).$$

The right-hand side is independent of $\mathcal{Q}$ then the union on each $\mathcal{Q}$ in $\Psi'_{Hg,sH}$ of sets on the left-hand side proves the searched inclusion.

To prove the opposite inclusion, for each $h$ define $\mathcal{P}_h \in \Psi'_{Hg,sH}$ as the partition containing the sets $\{ghts, gts\}$ for each $t \in H$ and the remaining singletons. Then, note that, by Lemma 3,

$$\bigcap_{\{ghts,gts\} \in \mathcal{P}_h} \mathcal{I}(M_{ghts,gts}) = \mathcal{I}(M_{h,e}).$$

Hence,

$$\bigcup_{h \in H} \mathcal{I}(M_{h,e}) = \bigcup_{h \in H} \bigcap_{\{ghts,gts\} \in \mathcal{P}_h} \mathcal{I}(M_{ghts,gts}) \subseteq \bigcup_{\mathcal{Q} \in \Psi'_{Hg,sH}} \bigcap_{\{u,v\} \in \mathcal{Q}} \mathcal{I}(M_{u,v}).$$

This concludes the proof.

$\square$

## C   IMPLICATIONS ON PRACTICAL MODELS

**Lemma 4.** *An element $(\alpha, \beta) \in \rho(1\text{-CNN})$ if and only if there exist a permutation of $\sigma \in S_n$ such that $\alpha_i = \beta_{\sigma(i)}$ for each $i = 1, \ldots, n$.*

*Proof.* Write $[n] \sqcup [n]' = \{1, \ldots, n, 1', \ldots, n'\}$, and notice that $\mathrm{Aff}_{\mathbb{Z}_n}(\mathbb{R}^n, \mathbb{R})' = \langle \mathbb{1}_{[n]} - \mathbb{1}_{[n]'} \rangle$, hence $\Psi'$ as defined in Remark 8 is composed by partitions $\mathcal{Q}$ of $[n] \sqcup [n]'$ such that

$$\mathcal{Q} = \{\{i, j'\} \mid i \in [n], j \in [n]'\}.$$

Recall $M^1 = \langle id_{\mathbb{R}^n \oplus \mathbb{R}^{n'}} \rangle$. Note that $(\alpha, \beta) \in \mathcal{I}(M^1_{i,j'}) = \langle id_{\mathbb{R}^n \oplus \mathbb{R}^{n'}} \rangle$ if and only if $\alpha_i = \beta_j$. Moreover, for a given $\mathcal{Q}$ in $\Psi'$, we have $(\alpha, \beta) \in \bigcap_{i,j' \in \mathcal{Q}} \mathcal{I}(M^1_{i,j'})$ if and only if, given the bijection $\sigma : [n] \to [n]'$ associating $i$ to $j'$, $\alpha_i = \beta_{\sigma(i)}$ for each $i = 1, \ldots, n$.

Notice that, by Theorem 1,

$$(\alpha, \beta) \in \rho(\mathcal{N}_\sigma(M^1, \mathrm{Aff}_{\mathbb{Z}_n}(\mathbb{R}^n, \mathbb{R})) = \bigcup_{\mathcal{Q} \in \Psi} \bigcap_{i,j' \in \mathcal{Q}} \mathcal{I}(M^1_{i,j'}),$$

which is equivalent at saying that there exist a permutation of $\sigma \in S_n$ such that $\alpha_i = \beta_{\sigma(i)}$ for each $i = 1, \ldots, n$. $\square$

*Proof of Proposition 6.* Note that $\rho(1\text{-CNN})$ is characterized by Lemma 4 as follows: $(\alpha, \beta) \in \rho(1\text{-CNN})$ if and only if there exists a permutation $\sigma \in S_n$ such that $\alpha_i = \beta_{\sigma(i)}$ for each $i = 1, \ldots, n$. In contrast, Proposition 4 shows that $(\alpha, \beta) \in \rho(\mathcal{N}_\sigma(M^n, \mathrm{Aff}_{\mathbb{Z}_n}(\mathbb{R}^n, \mathbb{R})))$ if and only if there exists an element $g \in \mathbb{Z}_n$ such that $\alpha_i = \beta_{i+g \pmod n}$ for each $i = 1, \ldots, n$. Notice that for $n > 1$, $\mathbb{Z}_n \lneq S_n$, hence $\rho(n\text{-CNN}) \subsetneq \rho(1\text{-CNN})$, as desired. The proof of the chain of inclusions

$$\rho(n\text{-CNN}) \subseteq \cdots \subseteq \rho(2\text{-CNN}) \subseteq \rho(1\text{-CNN})$$

is a direct consequence of Lemma 5 since: 1-CNN $\subseteq$ 2-CNN $\subseteq \cdots \subseteq$ n-CNN. $\square$

## D  FUNCTIONAL EQUATIONS

In this section, we introduce key results from the theory of functional equations that are necessary to prove Theorem 1. A functional equation, by definition, is an identity involving unknown functions as variables, and common examples include differential and integral equations (Kannappan, 2009). Here, we are particularly interested in the class of linear functional equations, which we explore in greater detail in the following section.

### D.1  LINEAR FUNCTIONAL EQUATIONS

Linear functions equations are functional equations which, for given $a_i \in \mathbb{R}$ and $b_i \in \mathbb{R}^d$, are defined by

$$\sum_{i=1}^{n} a_i \sigma(b_i x) = 0 \qquad (\forall x \in \mathbb{R}^d).$$

In particular, Theorem 9 is a fundamental tool in the proof of Theorem 1, since it characterizes the set of parameters $b_1, \ldots, b_n$ for which the specific case of linear functional equation in (22) is always satisfied for a non-polynomial $\sigma$ and arbitrary $a_1, \ldots, a_n \in \mathbb{R}$.

**Theorem 9.** *Let $\sigma : \mathbb{R} \to \mathbb{R}$ be a non-polynomial continuous function and $a_1, \ldots, a_n \in \mathbb{R}$. Let $\mathcal{P}$ be a partition of $[n]$ and define*

$$\Psi = \{\mathcal{Q} \le \mathcal{P} \mid \sum_{i \in P} a_i = 0 \; \forall P \in \mathcal{Q}\}.$$

*The set $B$ of elements $b = (b_1, \ldots, b_n) \in \mathbb{R}^{n \times m}$ which satisfy*

$$\sum_{P \in \mathcal{P}} \sum_{i \in P} a_i \sigma\Big(b_i \cdot x + y_P\Big) = 0 \qquad \Big(\forall x \in \mathbb{R}^m \forall y = (y_P)_{P \in \mathcal{P}} \in \mathbb{R}^{\mathcal{P}}\Big) \tag{22}$$

*is*

$$\bigcup_{\mathcal{Q} \in \Psi} \{(b_1, \ldots, b_n) \mid b_{i_1} = \cdots = b_{i_k} \; \forall \{i_1, \ldots, i_k\} \in \mathcal{Q}\}. \tag{23}$$

*Equivalently,*

$$B = \bigcup_{\mathcal{Q} \in \Psi} \bigcap_{\substack{P \in \mathcal{Q} \\ i,j \in P}} \{(b_1, \ldots, b_n) \mid b_i - b_j = 0\}.$$

*For arbitrary continuous functions $\sigma$, it is only true that the set defined in* (23) *is contained in $B$.*

To prove Theorem 9, we first need to prove some auxiliary results. Theorem 10, stated below, is a reformulation of Theorem 2.27 in Kiss & Laczkovich (2014) adapted here to the context of continuous real functions for convenience. For further discussion, refer to Appendix D.2.

**Theorem 10.** *Let $a_1, \ldots, a_n$ non-null real values, and let $b_1, \ldots, b_n \in \mathbb{R}^m$ be distinct real vectors. Continuous solutions $\sigma : \mathbb{R} \to \mathbb{R}$ of*

$$\sum_i a_i \sigma\Big(b_i \cdot x + y\Big) = 0 \qquad (\forall x \in \mathbb{R}^m \forall y \in \mathbb{R}) \tag{24}$$

*are polynomial.*

Moreover, to prove Theorem 9, the following notions and auxiliary results are required.

**Definition 14.** *Let $b = (b_1, \ldots, b_n) \in \mathbb{R}^{n \times m}$, the* identity pattern *of $b$ is the coarser partition $\mathcal{P}$ of $[n]$ such that $b_i = b_j$ for each $i, j \in P$ and $P \in \mathcal{P}$.*

**Theorem 11.** *Let $\sigma : \mathbb{R} \to \mathbb{R}$ be a non-polynomial continuous function, and $a_1, \ldots, a_n \in \mathbb{R}$. Then $b = (b_1, \ldots, b_n) \in \mathbb{R}^{n \times m}$ satisfies*

$$\sum_{i=1}^{n} a_i \sigma\Big(b_i \cdot x + y\Big) = 0 \qquad (\forall x \in \mathbb{R}^m \forall y \in \mathbb{R}) \tag{25}$$

*if and only if $\sum_{i \in P} a_i = 0$ for each $P$ in the identity pattern of $b$.*

*Proof.* Let $P_1, \ldots, P_q$ be the parts in the identity pattern of $b$ such that $\sum_{i \in P_j} a_i \neq 0$, define $a'_j = \sum_{i \in P_j} a_i$ then we can rewrite the equation in (25) as

$$\sum_{j=1}^{q} a'_j \sigma \left( b'_j \cdot x + y \right) = 0,$$

where, for each $j = 1, \ldots, q$, the value of $b'_j$ is set to the value of the $b_i$s for $i \in P_j$, which are all equal to each other. Since the $a'_j$ are non-null and $b'_j$ are distinct, by Theorem 10, $\sigma$ have to be polynomial which is impossible. To prove the opposite implication, let $\mathcal{P}$ be the identity pattern of $b$ and write

$$\sum_{i=1}^{n} a_i \sigma \left( b_i \cdot x + y \right) = \sum_{P \in \mathcal{P}} \sum_{i \in P} a_i \sigma \left( b_i \cdot x + y \right) = \sum_{P \in \mathcal{P}} \left( \sum_{i \in P} a_i \right) \sigma \left( b_i \cdot x + y \right),$$

where the last equality is possible because $b_i = b_j$ for each $i, j \in P$. $\qquad\square$

*Remark* 7. Note that the second implication of Theorem 11 holds for any $\sigma$, including polynomial functions.

This theorem gives the following corollary, which is the one actually needed to prove Theorem 9.

**Corollary 2.** *Let $\sigma : \mathbb{R} \to \mathbb{R}$ be a non-polynomial continuous function and $a_1, \ldots, a_n \in \mathbb{R}$. Let $\mathcal{P}_n$ be the set of all partition of $[n]$ and define*

$$\Phi = \{ \mathcal{P} \in \mathcal{P}_n \mid \sum_{i \in P} a_i = 0 \; \forall P \in \mathcal{P} \}.$$

*The set $B$ of elements $b = (b_1, \ldots, b_n) \in \mathbb{R}^{n \times m}$ satisfying (25) is*

$$\bigcup_{\mathcal{P} \in \Phi} \{ (b_1, \ldots, b_n) \mid b_{i_1} = \cdots = b_{i_k} \; \forall \{i_1, \ldots, i_k\} \in \mathcal{P} \}. \tag{26}$$

*Or equivalently,*

$$B = \bigcup_{\mathcal{P} \in \Phi} \bigcap_{\substack{P \in \mathcal{P} \\ i,j \in P}} \{ (b_1, \ldots, b_n) \mid b_i - b_j = 0 \}.$$

*For arbitrary continuous functions $\sigma$, it is only true that the set defined in (26) is contained in $B$.*

*Proof.* Define

$$B' = \bigcup_{\mathcal{P} \in \Phi} \{ (b_1, \ldots, b_n) \mid b_{i_1} = \cdots = b_{i_k} \; \forall \{i_1, \ldots, i_k\} \in \mathcal{P} \}.$$

By Theorem 11, $b$ satisfies (25) if and only if $\sum_{i \in P} a_i = 0$ for each $P$ in the identity pattern of $b$. Thus, $B \subseteq B'$. To prove the opposite inclusion, note that if $b = (b_1, \ldots, b_n) \in B'$ then there exist $\mathcal{P} \in \Phi$ such that $b$ has identity pattern $\mathcal{P}$, then, as $\sum_{i \in P} a_i = 0$ for each $P \in \mathcal{P}$, (25) is verified. Finally, note that this implication holds for any $\sigma$ by Remark 7, proving the last claim in Corollary 2. $\qquad\square$

*Proof of Theorem 9.* Notice that the problem

$$\sum_{P \in \mathcal{P}} \sum_{i \in P} a_i \sigma \left( b_i \cdot x + y_P \right) = 0 \qquad \left( \forall x \in \mathbb{R}^m \forall y = (y_P)_{P \in \mathcal{P}} \in \mathbb{R}^{\mathcal{P}} \right) \tag{27}$$

is equivalent to

$$\sum_{P \in \mathcal{P}} \sum_{i \in P} a_i \sigma \left( b_i \cdot x + y_P + \hat{y} \right) = 0 \quad \left( \forall x \in \mathbb{R}^m \forall y \in \mathbb{R}^{\mathcal{P}} \forall \hat{y} \in \mathbb{R} \right)$$

through the change of variables $y_P \mapsto y_P + \hat{y}$ for each $P \in \mathcal{P}$. This problem is in turn equivalent to

$$\sum_{i=1}^{n} a_i \sigma \left( \hat{b}_i \cdot \hat{x} + \hat{y} \right) = 0 \qquad \left( \forall \hat{x} \in \mathbb{R}^m \oplus \mathbb{R}^{\mathcal{P}} \forall \hat{y} \in \mathbb{R} \right) \tag{28}$$

due to the following change of variables

$$\hat{b}_i \mapsto \begin{pmatrix} b_i \\ e_P \end{pmatrix} \text{ for } i \in P, \text{ and } \hat{x} \mapsto \begin{pmatrix} x \\ y_P e_P \end{pmatrix},$$

where $\{e_P\}_{P \in \mathcal{P}}$ is the canonical base of $\mathbb{R}^{\mathcal{P}}$. Corollary 2 implies that the solution to (28) is

$$\bigcup_{\mathcal{Q} \in \Phi} \{(\hat{b}_1, \ldots, \hat{b}_n) \mid \hat{b}_{i_1} = \cdots = \hat{b}_{i_k} \ \forall \{i_1, \ldots, i_k\} \in \mathcal{Q}\}. \tag{29}$$

Note that $\hat{b}_{i_1} = \cdots = \hat{b}_{i_k}$ if and only if $b_{i_1} = \cdots = b_{i_k}$ and $\{i_1, \ldots, i_k\} \subseteq P$ for some $P \in \mathcal{P}$ if and only if $b_{i_1} = \cdots = b_{i_k}$ and $\{i_1, \ldots, i_k\} \in \mathcal{Q}$ for some $\mathcal{Q} \leq \mathcal{P}$.

Recall the definitions

$$\Phi = \{\mathcal{Q} \in \mathcal{P}_n \mid \sum_{i \in P} a_i = 0 \ \forall P \in \mathcal{Q}\} \text{ and } \Psi = \{\mathcal{Q} \leq \mathcal{P} \mid \sum_{i \in P} a_i = 0 \ \forall P \in \mathcal{Q}\}.$$

Noting that $\Psi = \{\mathcal{Q} \in \Phi \mid \mathcal{Q} \leq \mathcal{P}\}$, equation (29) implies that the solutions of (27) are

$$\bigcup_{\mathcal{Q} \in \Psi} \{(b_1, \ldots, b_n) \mid b_{i_1} = \cdots = b_{i_k} \ \forall \{i_1, \ldots, i_k\} \in \mathcal{Q}\}.$$

The final claim follows directly from the concluding statement in Corollary 2. □

*Remark* 8. In Theorem 9 the union

$$\bigcup_{\mathcal{Q} \in \Psi} \{(b_1, \ldots, b_n) \mid b_{i_1} = \cdots = b_{i_k} \ \forall \{i_1, \ldots, i_k\} \in \mathcal{Q}\} \tag{30}$$

has redundancies. Indeed, $\{(b_1, \ldots, b_n) \mid b_{i_1} = \cdots = b_{i_k} \ \forall \{i_1, \ldots, i_k\} \in \mathcal{Q}\}$ is contained in $\{(b_1, \ldots, b_n) \mid b_{i_1} = \cdots = b_{i_k} \ \forall \{i_1, \ldots, i_k\} \in \mathcal{R}\}$ for each $\mathcal{Q} \leq \mathcal{R}$ finer partitions of $\mathcal{P}$. Hence, the set defined by (30) is the same as

$$\bigcup_{\mathcal{Q} \in \Psi'} \{(b_1, \ldots, b_n) \mid b_{i_1} = \cdots = b_{i_k} \ \forall \{i_1, \ldots, i_k\} \in \mathcal{Q}\}$$

where $\Psi'$ is the subset of $\Psi$ containing only the minimal partitions with respect to the refinement ordering.

## D.2 GENERALIZED POLYNOMIALS IN THE CONTINUOUS CASE

In Appendix D.1, we employ Theorem 10, a reformulation of Theorem 2.27 in Kiss & Laczkovich (2014), adapted here for convenience to the context of continuous real functions. In particular, the original version of this theorem proves that arbitrary complex functions satisfying (24) are generalized polynomials, defined as follows.

**Definition 15.** *A function $\sigma : \mathbb{C} \to \mathbb{C}$ is a* generalized monomial *function if there exist a symmetric function $F : \mathbb{C}^n \to \mathbb{C}$ additive in each of its variables, such that $\sigma(x) = F(x, \ldots, x)$ for each $x \in \mathbb{C}$. A function $\sigma : \mathbb{C} \to \mathbb{C}$ is a* generalized polynomial *if it is a finite sum of generalized monomials, we say that a generalized polynomial $\sigma$ is* real *if $\sigma$ is real and there exists a symmetric function $F : \mathbb{C}^n \to \mathbb{R}$ additive in each of its variables, such that $\sigma(x) = F(x, \ldots, x)$ for each $x \in \mathbb{R}$.*

Trivially, since complex functions satisfying (24) are generalized polynomials, any real solutions are real generalized polynomials.

To conclude the proof of Theorem 10, it remains to show that continuous real generalized polynomials are simply real polynomial functions, as shown by Proposition 17.

**Proposition 17.** *A real continuous generalized polynomial is a real polynomial function.*

*Proof.* The proof of Theorem 17 will be analogous to the proof of the classical proof that any continuous real additive function is linear, see Theorem 1.1 in Kannappan (2009).

First, we show that real generalized monomials are monomial functions on rational numbers.

Indeed, suppose first that $f$ is a real generalized monomial and let $F : \mathbb{R}^n \to \mathbb{R}$ be the symmetric function additive in each variable and such that $f(x) = F(x, \dots, x)$ for each $x \in \mathbb{R}$. Note that for each $r \in \mathbb{N}$,

$$F(x_1, \dots, rx_i, \dots, x_n) = F(x_1, \dots, x_i + \cdots + x_i, \dots, x_n) =$$
$$F(x_1, \dots, x_i, \dots, x_n) + \cdots + F(x_1, \dots, x_i, \dots, x_n) = rF(x_1, \dots, x_i, \dots, x_n). \quad (31)$$

Note that $F(x_1, \dots, x_{i-1}, 0, x_{i+1}, \dots, x_n) = 0$, indeed

$$F(x_1, \dots, x_{i-1}, 0, x_{i+1}, \dots, x_n) =$$
$$F(x_1, \dots, x_{i-1}, 0 + 0, x_{i+1}, \dots, x_n) = \quad (32)$$
$$F(x_1, \dots, x_{i-1}, 0, x_{i+1}, \dots, x_n) + F(x_1, \dots, x_{i-1}, 0, x_{i+1}, \dots, x_n)$$

Eliminating a term $F(x_1, \dots, x_{i-1}, 0, x_{i+1}, \dots, x_n)$ from both the sides of (32), we get the required result.

Furthermore, $F(x_1, \dots, x_i, \dots, x_n) = -F(x_1, \dots, -x_i, \dots, x_n)$. Indeed,

$$F(x_1, \dots, x_i, \dots, x_n) + F(x_1, \dots, -x_i, \dots, x_n) = F(x_1, \dots, x_i - x_i, \dots, x_n) = 0. \quad (33)$$

Equations (31) and (33) yields

$$F(x_1, \dots, rx_i, \dots, x_n) = rF(x_1, \dots, x_i, \dots, x_n) \quad (34)$$

for each $r \in \mathbb{Z}$. Note that by substituting $rx_i = y_i$, we obtain

$$F(x_1, \dots, y_i, \dots, x_n) = rF(x_1, \dots, \frac{1}{r}y_i, \dots, x_n)$$

Equivalently,

$$F(x_1, \dots, \frac{1}{r}y_i, \dots, x_n) = \frac{1}{r}F(x_1, \dots, y_i, \dots, x_n) \quad (35)$$

Equations (34) and (35) prove

$$F(x, \dots, rx, \dots, x) = rF(x, \dots, x) \quad (36)$$

for each $r \in \mathbb{Q}$. Hence,

$$f(rx) = F(rx, \dots, rx) = r^n F(x, \dots, x) = r^n f(x).$$

for each $r \in \mathbb{Q}$. In particular set $x = 1$ and $f(1) = c \in \mathbb{R}$,

$$f(r) = r^n f(1) = cr^n.$$

Hence, a real generalized monomial is a monomial on $\mathbb{Q}$.

Finally, we can prove the general case where $f$ is a real generalized polynomial. Recalling that real generalized polynomials are sums of real generalized monomials, they are sums of real monomial functions on $\mathbb{Q}$, namely polynomial functions on $\mathbb{Q}$.

We conclude by noting that, since $f$ is continuous, it extends as a polynomial function on $\mathbb{R}$ due to continuity. $\qquad \square$

## E  TECHNICAL LEMMAS

In what follows, let $\mathcal{C}$, $\mathcal{D}$ and $\mathcal{F}$ be families of functions in $\mathcal{C}(X, V)$, where $X$ is a topological space and $V$ a real vector space.

**Lemma 5.** *If $\mathcal{C} \subseteq \mathcal{D}$, then $\rho(\mathcal{D}) \subseteq \rho(\mathcal{C})$.*

**Lemma 6.** *Let $\mathcal{C}$ and $\mathcal{D}$ be two families of real-valued functions such that each of them contains at least a constant function. The equivalence relations induced by their identification condition are linked by the following conditions $\rho(\mathcal{C} + \mathcal{D}) = \rho(\mathcal{C} \cup \mathcal{D}) = \rho(\mathcal{C}) \cap \rho(\mathcal{D})$.*

*Proof.* Let us prove the first equality. Let $c$ be the constant function in $\mathcal{D}$. Hence $\rho(\mathcal{C} + \mathcal{D}) \subseteq \rho(\mathcal{C} + c) = \rho(\mathcal{C}) \subseteq \rho(\mathcal{C}) \cup \rho(\mathcal{D})$. To prove the inverse inclusion, suppose there exists a function $f$ either in $\mathcal{C}$ or $\mathcal{D}$ separating $x$ and $y$. Without loss of generality, suppose $f \in \mathcal{C}$, $f + c \in \mathcal{C} + \mathcal{D}$ would be separating $x$ and $y$. This conclude the proof of the first equality. The proof of the second equality follows from the definition of $\rho$. Indeed,

$$\rho(\mathcal{C} \cup \mathcal{D}) = \{(x, y) \in X \times X \mid f(x) = f(y) \, \forall f \in \mathcal{C} \cup \mathcal{D}\} =$$
$$\{(x, y) \in X \times X \mid f(x) = f(y) \, \forall f \in \mathcal{C}\} \cap \{(x, y) \in X \times X \mid f(x) = f(y) \, \forall f \in \mathcal{D}\} =$$
$$\rho(\mathcal{C}) \cap \rho(\mathcal{D}).$$

$\square$

*Remark* 9. Note that, with slight modifications to the proofs, analogous results to all previous lemmas can be derived by substituting $\rho$ with $\mathcal{I}$.

**Lemma 7.** *If $\mathcal{F}_d$ is a set spanning a null-bias space $M_d$, then*

$$\mathcal{I}(\mathcal{N}_\sigma(M_1, \ldots, M_d)) = \mathcal{I}(\mathcal{N}_\sigma(M_1, \ldots, M_{d-1}, \mathcal{F}_d)).$$

*Proof.* Trivially,
$$\mathcal{N}_\sigma(M_1, \ldots, M_{d-1}, \mathcal{F}_d)) \subseteq \mathcal{N}_\sigma(M_1, \ldots, M_d)).$$

For the zero-locus analogous of Lemma 5,

$$\mathcal{I}(\mathcal{N}_\sigma(M_1, \ldots, M_d)) \subseteq \mathcal{I}(\mathcal{N}_\sigma(M_1, \ldots, M_{d-1}, \mathcal{F}_d)).$$

To prove the opposite inclusion, write $\mathcal{F}_d = \{\phi_1, \ldots, \phi_s\}$ and note that each neural network $\eta^d$ in $\mathcal{N}_\sigma(M_1, \ldots, M_d)$ can be written as

$$\eta^d = (x_1 \phi_1 + \cdots + x_s \phi_s) \circ \tilde{\sigma} \circ \eta^{d-1} = x_1(\phi_1 \circ \tilde{\sigma} \circ \eta^{d-1}) + \cdots + x_s(\phi_s \circ \tilde{\sigma} \circ \eta^{d-1}),$$

for some $x_1, \ldots, x_s \in \mathbb{R}$ and $\eta^{d-1} \in \mathcal{N}_\sigma(M_1, \ldots, M_{d-1})$.

Moreover, note that
$$\phi_i \circ \tilde{\sigma} \circ \eta^{d-1} \in \mathcal{N}_\sigma(M_1, \ldots, M_{d-1}, \mathcal{F}_d),$$

for each $i = 1, \ldots, s$.

If $\beta \in \mathcal{I}(\mathcal{N}_\sigma(M_1, \ldots, M_{d-1}, \mathcal{F}_d))$, then

$$\eta^d(\beta) = x_1(\phi_1 \circ \tilde{\sigma} \circ \eta^{d-1}) + \cdots + x_s(\phi_s \circ \tilde{\sigma} \circ \eta^{d-1}) = 0.$$

Thus,
$$\mathcal{I}(\mathcal{N}_\sigma(M_1, \ldots, M_{d-1}, \mathcal{F}_d)) \subseteq \mathcal{I}(\mathcal{N}_\sigma(M_1, \ldots, M_d)),$$

completing the proof. $\square$

