# OpenReview forum: "Separation Power of Equivariant Neural Networks"
_ICLR.cc/2025/Conference — ICLR 2025 Poster_

### Official Review · Reviewer_ESKT · 2024-11-03

**Soundness:** 3
**Presentation:** 2
**Contribution:** 3
**Rating:** 6
**Confidence:** 3

**Summary:**

The paper proves a number of results about the separation power of equivariant networks. In particular, it focusses on networks equivariant to regular / permutation representations of finite groups. The authors give a recursive formula for the set of pairs of inputs that are indistinguishable by any function in the function class under consideration. Further theorems clarify the role of non-linearities (they don't matter, provided they are non-polynomial), depth (increases separation power, but not indefinitely), multiplicity and invariant features (don't affect separability). Finally, the paper discussed implications for IGNs and CNNs.

**Strengths:**

+ The paper provides a detailed characterization of input separability for regular equivariant networks.
+ The theorems shed light on the factors that do and do not affect separability, which is quite insightful.
+ The theorems seem non-trivial and novel

**Weaknesses:**

- The paper is rather long-winded, with results starting only on page 7.
- The "twin network trick" is presented as a key insight, but it seems like a rather trivial observation to me. Eta(a) = eta(b) iff eta(a) - eta(b) = 0.
- Not much intuition is provided, e.g. for theorem 1, which I found hard to understand or see why it must be true. Proofs are all in the appendix. It would be much better to move some background material to the appendix and explain the intuition behind the theorems and perhaps a proof sketch in the main paper.

**Questions:**

Typo: Line 162 has a typo; "ING" should be "IGN."

---

> ### Author Response · Authors · 2024-11-21
> **Response to Reviewer ESKT**
>
> We are thankful for the reviewer’s positive assessment and especially value their constructive comments and recommendations. In the following, we respond to each of their questions and concerns.
>
> **Clarity and organization:**
> > The paper is rather long-winded, with results starting only on page 7.
>
> > Not much intuition is provided, e.g. for theorem 1, which I found hard to understand or see why it must be true. Proofs are all in the appendix. It would be much better to move some background material to the appendix and explain the intuition behind the theorems and perhaps a proof sketch in the main paper.
>
> We agree with the reviewer that the organization of the material required some updates. In particular, in the revised version of the manuscript, we have:
>
> - Streamlined preliminary material in Section 4.1 and referred to Appendix for better explanation of the topic.
> - Removed Example 1, as it was generalized by Example 2, and streamlined Example 4 from Section 4.2 due to similarities with Example 3 (following the notation of the originally submitted manuscript).
> - We have reorganized the presentation of Theorem 1 by introducing and recalling the necessary notation before the theorem statement.
> - In the updated manuscript, we emphasized the significance of Theorem 1 as a fundamental tool for proving nearly all subsequent results, including Theorems 2, 3, 4, and Proposition 4.
> - As suggested by the reviewer, we have added proof sketches for Theorems 2, 3, and 5. Notably, Theorem 2 serves as an illustrative example of the application of Theorem 1, while Theorem 3 demonstrates the combined application of both Theorems 1 and 2. Still, we opted not to include proof sketches for Theorems 1 and 4, since this would have negatively impacted readability, due to the technical nature of the proofs and the additional required notation.
>
> **Relevance of the twin network trick:**
>
> > The "twin network trick" is presented as a key insight, but it seems like a rather trivial observation to me. $\eta(a) = \eta(b)$ iff $\eta(a) - \eta(b)$ = 0.
>
> We agree that the trick in itself is rather simple, but its use if fundamental to develop the following theory. To this end, it is crucial to describe how the neural space transforms under this trick, in particular by describing how the architecture changes under this operation (eq. (4) and the following discussion). Moreover, Section 5.1 is important to layout the notation and justification for moving from a separation problem to an identification problem.
>
> **Other:**
>
> > Typo: Line 162 has a typo; "ING" should be "IGN."
>
> Thank you for the careful reading. We fixed this issue.

---

> > ### Comment · Reviewer_ESKT · 2024-11-22
> > **Update**
> >
> > Thanks for the response. I will keep my score as is.

---

### Official Review · Reviewer_ceR4 · 2024-11-03

**Soundness:** 3
**Presentation:** 3
**Contribution:** 3
**Rating:** 8
**Confidence:** 3

**Summary:**

This work analyzes the separability of deep learning models with equivariant architectures (that is, architectures that are equivariant to specific group actions on the input and output spaces). Separability in this case means identifying pairs of points that take identical values under all networks of a fixed architecture. The work only considers separability of pairs of points. The paper makes a few assumptions on the network, the most notable being that the network uses a permutation representation (and hence the group $G$ is assumed to be finite). After setting up notation related to equivariant architectures, the work describes the “separation trick”, which essentially shifts perspective from looking for points $\alpha, \beta$ where $f(\alpha) = f(\beta)$ for all $f$ in the function class and instead looks at pairs of points $\alpha, \beta$ where $\eta(\alpha,\beta) = f(\alpha) – f(\beta) = 0$. That is, the zero locus of $\eta$. The paper proves a recursive formula about separability within this framework and then draws some conclusions about how various architecture choices impact separability (e.g., depth, width, etc.).

**Strengths:**

**Clarity:** The work is clearly written and relatively crisp. The Preliminaries section does a decent job introducing the basic tools needed in the paper (e.g., the notion of equivariance, etc.). An explanation of how the work fits into the current research landscape is provided. Despite being a theory paper, this work should be accessible to a wide range of readers.

**Problem importance:** While there is now a substantial body of work on equivariant architectures, both in terms of architectures for specific symmetry groups and applications, and theory, significant questions remain. Gaining a better mathematical understanding of what we sacrifice by imposing equivariance in our models is an interesting and important question that deserves more research.

**Weaknesses:**

**Theorem 1 is hard to read:** As this is one of the critical contributions of the work, it would be worth polishing it to make it more readily understandable, especially since the work takes pains in Section 4 to layout the framework for this result. It feels like in the process of trying to make the theorem more “informal” all the explanations were taken out but the mathematical notation was retained. For instance, what is the symbol “\leq” at line 377? The reviewer can make a guess, but it would be good to state this explicitly. Similarly, there a 6 different indices in (5). Is it possible to either break up this expression or build up to it? It is also a little hard to read since 4 of these indices ($h$, $k$, $\mathcal{Q}$, and $P$) don’t appear in the argument. This reviewer’s advise would be to cut down on some of the lengthy preliminaries and devote a bit more time to setting up and describing Theorem 1. It was hard for this reviewer to draw much insight from this Theorem.

**CNNs are equivariant to integer translation, $\mathbb{Z}^2$, not cyclic translation:** As far as this reviewer is aware, it is relatively uncommon to use circular padding in computer vision. For this reason, most CNNs are not equivariant to a product of finite cyclic groups, but rather to $\mathbb{Z}^2$. It would be worth stating that Example 4 does not represent the majority of CNNs.

**Hints at proof strategy:** While the reviewer appreciates that space constraints mean that the proofs need to be moved to the supplementary material, it would make the work more valuable if a high-level description of how the results were proven were included. After all, it is sometimes the case that the insights in the proof are more valuable than the theorems themselves. Again, the reviewer would advise shortening the preliminary material to provide more space. For instance, the preliminary material discusses group representations but (to this reviewer’s memory) the only place this later appears is a reference to the regular representation.

### Nitpicks:

-	Line 161: “From a practical viewpoint is fundamental to understand which are the hyperparameters …” $\mapsto$ “From a practical viewpoint it is fundamental to understand the hyperparameters…”

**Questions:**

- The reviewer was surprised that despite this being a result about equivariant architectures, group and representation-theoretic machinery featured very little (acknowledging that the reviewer did not go through the supplementary material closely). Is this the case or did the reviewer just miss something?
- Have the authors thought about separability of more than pairs of points? Can anything be said about this?

---

> ### Author Response · Authors · 2024-11-21
> **Response to Reviewer ceR4 - Part 1/2**
>
> We appreciate the reviewer’s positive feedback, particularly the helpful suggestions and insights they shared. Below, we address their specific questions and concerns.
>
> **Theorem 1 is hard to read:**
>
> > As this is one of the critical contributions of the work, it would be worth polishing it to make it more readily understandable, especially since the work takes pains in Section 4 to layout the framework for this result. It feels like in the process of trying to make the theorem more “informal” all the explanations were taken out but the mathematical notation was retained. For instance, what is the symbol “\leq” at line 377? The reviewer can make a guess, but it would be good to state this explicitly. Similarly, there a 6 different indices in (5). Is it possible to either break up this expression or build up to it? It is also a little hard to read since 4 of these indices ($h$, $k$, $\mathcal{Q}$, and $P$) don’t appear in the argument. This reviewer’s advise would be to cut down on some of the lengthy preliminaries and devote a bit more time to setting up and describing Theorem 1. It was hard for this reviewer to draw much insight from this Theorem.
>
> We agree with the reviewer’s observation and we addressed this issue in the updated manuscript as follows:
>
> 1. **Readability:** We revised the presentation of Theorem 1 by defining and recalling the necessary notation before stating the theorem. Furthermore, we ensured that the theorem and its preamble are presented on the same page, improving both clarity and readability.
> 3. **Insight on Theorem 1:** Additionally, the reviewer’s observation made us realize that, while we emphasized this theorem’s importance in the manuscript, the reasons for its significance may not be immediately clear. We addressed this issue by focusing on the following weakness highlighted by the reviewer.
>
> **Hints at proof strategy:**
>
> > While the reviewer appreciates that space constraints mean that the proofs need to be moved to the supplementary material, it would make the work more valuable if a high-level description of how the results were proven were included. After all, it is sometimes the case that the insights in the proof are more valuable than the theorems themselves.
>
> We moved less relevant preliminary notation to the appendix and included additional intuition on the techniques used to prove Theorems 2, 3, and 5. In the proof outlines for Theorems 2 and 3, we highlighted the pivotal role of Theorem 1, directly addressing the reviewer’s earlier observation.
>
> **CNNs are equivariant to integer translation, $\mathbb{Z}^2$, not cyclic translation:**
>
> > As far as this reviewer is aware, it is relatively uncommon to use circular padding in computer vision. For this reason, most CNNs are not equivariant to a product of finite cyclic groups, but rather to $\mathbb{Z}^2$. It would be worth stating that Example 4 does not represent the majority of CNNs.
>
> We agree with the reviewer that Example 4 (Example 3 in the revised version) does not represent the majority of CNNs. Accordingly, we have revised references to standard CNNs throughout the paper.
>
> However, in our view, practical CNNs are not strictly $\mathbb{Z}_n^2$ or $\mathbb{Z}^2$-equivariant; in the latter case, their filters and signals would need to be defined and convolved over the entire $\mathbb{Z}^2$. Nevertheless, we believe that both of these models provide useful theoretical frameworks for analyzing certain properties of practical CNNs.
>
> We would appreciate the reviewer’s opinion on this matter and whether they believe that circular convolutions can still offer insights into practical models.

---

> ### Author Response · Authors · 2024-11-21
> **Response to Reviewer ceR4 - Part 2/2**
>
> **The role of representation theory in the work:**
>
> > Again, the reviewer would advise shortening the preliminary material to provide more space. For instance, the preliminary material discusses group representations but (to this reviewer’s memory) the only place this later appears is a reference to the regular representation.
>
> As rightfully suggested by the reviewer, we streamlined the preliminaries on representation theory. However, a significant portion remains in the main text because, although representation theory is not directly employed until Section 5.6, it plays a fundamental role in Proposition 1 and Examples 2 and 3. These provide essential groundwork to motivate and introduce the tools required to state Theorem 1, particularly the role of partitions $\mathcal{P}$.
>
> > The reviewer was surprised that despite this being a result about equivariant architectures, group and representation-theoretic machinery featured very little (acknowledging that the reviewer did not go through the supplementary material closely). Is this the case or did the reviewer just miss something?
>
> A significant portion of this work—specifically Theorems 1, 2, 3, and 8 (in the appendix, as a more general version of Theorem 4)—focuses on the layer space $M$, which consists simply of vector subspaces of affine functions. This definition, along with the proofs of the preceding results, does not involve representation theory; however, this level of generality is essential, as our framework must include twin-layer spaces of affine spaces. For further details, please refer to Appendix B.1.
>
> In contrast, representation theory plays a significant role in the second part of the paper. The proofs of Theorem 5 and, in particular, Proposition 4, rely heavily on results from representation theory; indeed, Appendix A.5 is entirely dedicated to introducing and developing the representation-theoretic tools needed to prove Proposition 4.
>
> **Separability beyond pairs of points:**
>
> >  Have the authors thought about separability of more than pairs of points? Can anything be said about this?
>
> This is an interesting suggestion. As future work, we plan to derive generalization bounds for equivariant models. In this regard, studying the separability of more than just pairs of points could be a promising approach for addressing shattering problems and computing the VC dimension of these models. However, a direct extension of our results to this more general case seems non-trivial.
>
> **Other:**
>
> > Line 161: “From a practical viewpoint is fundamental to understand which are the hyperparameters …” $\mapsto$ “From a practical viewpoint it is fundamental to understand the hyperparameters…”
>
> We appreciate the careful review and we have made the necessary adjustments in the revised version of the paper.

---

> > ### Comment · Reviewer_ceR4 · 2024-11-23
> > **Response to the auth**
> >
> > We thank the authors for their clarifications and are happy with the proposed changes. We have raised our score to an 8.

---

### Official Review · Reviewer_rxAf · 2024-11-04

**Soundness:** 4
**Presentation:** 3
**Contribution:** 4
**Rating:** 8
**Confidence:** 3

**Summary:**

This paper develops a theoretical framework for analyzing the separation power of equivariant neural networks, such as convolutional and permutation-invariant networks. The authors present a complete characterization of inputs that are indistinguishable by models derived by a given architecture, and derive how separability is influenced by hyperparameters and architectural choices (activation functions, depth, hidden layer width, and representation types). Key findings include: all non-polynomial activations achieve maximum separation power, depth improves separation up to a threshold, adding invariant features to hidden representations does not impact separation power, and and block decomposition of hidden representations affects separability.

**Strengths:**

The strengths of the paper are numerous. The authors provide a novel theoretical framework for analyzing separation power in equivariant networks, creatively combining group theory and functional analysis. Particularly interesting is the twin network trick which transforms a network separation problem into a zero locus problem for neural networks, allowing the application of recursive techniques for solving zero locus problems. The authors also provide new insights into how architectural choices affect the expressivity of a model – this is extremely valuable for the community.


The authors have provided clear motivations for the paper, rigorous mathematical proofs for all results, and clearly stated assumptions as well as limitations. Authors not only provide theorem statements but also give the reader intuitive understanding. The paper also does a good job of building from first principles which makes the read even more enjoyable!

**Weaknesses:**

The authors clearly state the limitations of the work. An obvious weakness is the computational complexity, but the goal of this paper is to build theoretical frameworks which is a very challenging task. It is understandable that the computational complexity is not practical. Though efficient computational approximates could be explored.

The paper could also benefit from experiments showing the effect (or lack thereof) of depth on separation power, as well as the impact of different activation functions. But again, the paper is a theoretical one and does not need to show experiments to be able to stand on its own legs.

It could be interesting to see the effect of initialization in separation power.

**Questions:**

Given the superpolynomial complexity of the recursive formula, are there special cases where computation becomes tractable?
What are the trade-offs between separation power and other desirable properties?
Is there a connection between degree of polynomial and separation power?
Have you explored connections between separation power and generalization bounds?
Are there cases where maximum separation power might be undesirable?

---

> ### Author Response · Authors · 2024-11-21
> **Response to Reviewer rxAf - Part 1/2**
>
> We gratefully acknowledge the reviewer’s positive evaluation, we especially appreciate the thoughtful observations and advice provided. In the following, we respond to their questions and concerns.
>
> **Computational complexity of (5):**
>
> > The authors clearly state the limitations of the work. An obvious weakness is the computational complexity, but the goal of this paper is to build theoretical frameworks which is a very challenging task. It is understandable that the computational complexity is not practical. Though efficient computational approximates could be explored.
>
> We agree that the computational complexity of formula (5) may be prohibitive. However, as the reviewer correctly noted, the formula is not intended for explicit computation of the set $\mathcal{I}$ (this would be similar to having an explicit formula for finding WL-indistinguishable graphs for GNNs), but rather to derive several practical implications in Sections 5 and 6. Nonetheless, we acknowledge that alternative, efficient computational approaches would be valuable and could be explored in future work.
>
> > Given the superpolynomial complexity of the recursive formula, are there special cases where computation becomes tractable?
>
> Yes. For example, in the case of equivariant shallow networks with regular representations, we can optimize (5) by reducing the computation of $\rho$ to that presented in Proposition 4. However, depending on the group $H$ (e.g., $S_n$), the computations may still be intractable.
>
> **Experimental results:**
>
> >  The paper could also benefit from experiments showing the effect (or lack thereof) of depth on separation power, as well as the impact of different activation functions. But again, the paper is a theoretical one and does not need to show experiments to be able to stand on its own legs.
>
> We agree with the reviewer that experiments would strengthen the soundness of the presented work. However, separability cannot be directly controlled by the experimenter; it can only be influenced by adjusting other factors, such as hyperparameters or activations. These factors, however, affect not only separability but also, in different ways, other critical aspects such as approximation power, generalization, and trainability, all of which in turn impact empirical outcomes. Thus, designing experimental settings to isolate the impact of separation power is challenging, as it is difficult to disentangle the effects of hyperparameter changes on these other aspects.
>
> It is our intention to continue studying the effects of parameter choices on these additional components. Once we have achieved a clearer understanding of their impact, we believe it will be feasible to conduct a detailed and meaningful experimental study.
>
> In alignment with the final part of the reviewer’s comment, we believe our theoretically grounded contribution is an important step toward understanding the learning pipeline of equivariant networks, providing a foundation for future theoretical results that can be empirically and meaningfully validated.
>
> **Initialization:**
>
> > It could be interesting to see the effect of initialization in separation power.
>
> This is an interesting question; however, our analysis centers on the separation power of entire neural network spaces, examining its implications for approximation theory and, consequently, for assessing generalization capabilities. Eventually, evaluating the separation power of single functions, potentially under specific initialization protocols, could be essential in future research on the training dynamics of the network spaces discussed above.

---

> ### Author Response · Authors · 2024-11-21
> **Response to Reviewer rxAf - Part 2/2**
>
> **Trade-off and connections with separation power:**
>
> > What are the trade-offs between separation power and other desirable properties?
>
> This is another interesting question. We partially address this point in Section 5, where we focus on the trade-off between separation power and computational cost required by the model. Specifically, we show that certain costly modifications to the architecture have limited or no impact on separation power, while others (most notably, the representation type—see Theorem 5) increase it. We have added a comment regarding this point on lines 477–478.
> Beyond this, we believe that our work provides a useful tool for studying the approximation and generalization capabilities of equivariant networks. In this context, further insights into the trade-off between separation power and other properties are likely to emerge.
>
> > Is there a connection between degree of polynomial and separation power?
>
> As noted in Section 7, our results prove that all non-polynomial activations have equal and maximal separation power. However, our framework is unable to identify polynomial activations with suboptimal separation power, as it relies on non-constructive results from the theory of linear functional equations. Consequently, we are unable to determine whether the degree of the polynomial influences an activation’s separation power. We added a comment to clarify this point at the end of Section 5.3.
>
> > Have you explored connections between separation power and generalization bounds?
>
> Generalization and approximation bounds are very interesting aspects, and in fact they motivated this work, as discussed in Section 3.1 and Section 3.2. Indeed, we believe that understanding the structure and behavior of the separation power under architecture modifications is a necessary step to address generalization and approximation bounds. Moreover, extending the study to the separability of points beyond pairs (as suggested by Reviewer ceR4) could help in understanding generalization by solving shattering problems and computing the VC dimension of equivariant models.
>
> > Are there cases where maximum separation power might be undesirable?
>
> This point is interesting, and we believe that there are at least two relevant cases, related to the aspects discussed above:
>
> 1. Achieving maximal separation power may require intractably large representations (see Theorem 5).
> 2. In the related field of GNNs, it has been shown that the relationship between separation power and VC dimension—and thus generalization—can be complex (see, e.g., [1, 2]). An analogous behavior could exist for equivariant neural networks, though directly extending these ideas is challenging, as the underlying tools differ significantly and require representation theory.
>
> ---
> [1] B. J. Franks et al., Weisfeiler–Leman at the margin: When more expressivity matters, ICML 2024.
> [2] C. Morris et al., WL meet VC, ICML 2023.

---

> > ### Comment · Reviewer_rxAf · 2024-11-29
> > **Response to the Authors**
> >
> > We thank the authors for their thorough answer. All my questions and concerts have been addressed. I will keep my score of 8.

---

### Official Review · Reviewer_1QZn · 2024-11-04

**Soundness:** 3
**Presentation:** 3
**Contribution:** 2
**Rating:** 6
**Confidence:** 3

**Summary:**

This paper considers the separation power of equivariant networks. In the setting of permutation representations and element-wise nonlinearities, this paper charactierizes the set of identified points and then prove (1) all non-polynomial activations equivalently share the same and maximum separation power; (2) larger depth indicates better separation power; and (3) separation power is independent
of invariant hidden features.

**Strengths:**

* This paper is well-written and easy to follow.

* This paper considers an interesting and important problem in machine learning area and extends our understanding of separation power of specific equivariant architectures.

* This paper's definitions and statments are clear,  and its theoretical analysis is also solid.

**Weaknesses:**

* The theoretical results in paper are interesting but not so supervising, thus it seems unable to improve equivariant network design.

* The setting in this paper is somehow too specific (finite group, and feedforward architectures with element-wise activation if I understand correctly), which may limit its contribution.

More detail can be found in Questions.

**Questions:**

It is an interesting and solid paper to me. However, it may not have a significant contribution due to its limited setting. If I understand correctly, this paper considers only feedforward architectures with element-wise activations under finite groups. Moreover, many results seem not superising, which may weaken your contribution to help equivariant architecture design.

Feedforward networks with ReLU activations may only allow generalized permutation representations since equivariance properties require that the activation and group elements are commutative. If it is right and can be extened to general element-wise activations, it will enhance the soundness of your setting. What's more, the network with biases and input $x$ may be equivalent with that without baises and input $(x,1)$, which may further extend your results.

It may be not surprising to solve the identified set since this paper adopts element-wise activations and permutation representations. While all common activations seem to be monotonously increasing and bijective, the choice of activation would not be assumed to affect the separation power. Besides, the set $\rho(\mathcal{N})$ is assumed to have a lower bound. As a result, when we increase the depth to infinity, it is obvious that the identified set $\rho(\mathcal{N})$ would not change after a threshold of the depth. So a more explicit result would be more interesting.

Therefore, I think this paper is interesting that may help improve our understanding to equivariant networks but may have relatively limited contributions.

---

> ### Author Response · Authors · 2024-11-21
> **Response to Reviewer 1QZn - Part 1/2**
>
> We thank the reviewer for their positive assessment and, in particular, for the constructive observations and suggestions they provided.
>
> Here, we aim to address their questions and concerns:
>
> **Impact:**
>
> > [...] Moreover, many results seem not superising, which may weaken your contribution to help equivariant architecture design.
>
> To the best of our knowledge, no formalization or proof of the results in our paper has been published before. In particular, while it may be intuitively reasonable to expect some dependency of separability on the activations, depth, width, and representation type, it is non-obvious how these dependencies occur. To address this, we quantitatively define these relationships in Theorems 2–5, thereby providing concrete guidelines for selecting key parameters in the design of architectures with provable guarantees on separation power.
>
> Moreover, to the best of our knowledge, the techniques we employ—such as the use of functional equations and reduction to zero-locus problems—are original and may serve by themselves as useful tools for researchers in related fields.
>
> **Limited setting:**
>
> > [...] However, it may not have a significant contribution due to its limited setting. If I understand correctly, this paper considers only feedforward architectures with element-wise activations under finite groups.
>
> > Feedforward networks with ReLU activations may only allow generalized permutation representations since equivariance properties require that the activation and group elements are commutative. If it is right and can be extened to general element-wise activations, it will enhance the soundness of your setting.
>
> > It may be not surprising to solve the identified set since this paper adopts element-wise activations and permutation representations.
>
> We acknowledge this limitation but wish to emphasize that permutation representations constitute a significant portion of the equivariant deep learning literature. Furthermore, by Theorem 2 in [1], element-wise non-odd activations equivariant to compact topological groups can only admit representations isomorphic to permutation representations, which are precisely the type addressed in our work.
> Thus, we believe our approach encompasses a substantial portion of the literature on equivariant networks with point-wise activations.
>
> [1] Pacini et al., A Characterization Theorem for Equivariant Networks with Point-wise Activations, ICLR 2024
>
> > While all common activations seem to be monotonously increasing and bijective, the choice of activation would not be assumed to affect the separation power.
>
> This does not always hold; for example, ReLU is monotonically increasing but not bijective, as it maps the entire half-line of negative real numbers to zero. Interestingly, our findings indicate that even such non-bijective activations possess the same separation power as bijective ones. However, the result remains non-trivial even for bijective activations. While, as the reviewer observes, bijective activations alone have clearly maximal separation power, evaluating the interplay between equivariant linear layers and potentially bijective activations becomes challenging when they are composed sequentially, as in equivariant neural networks.
> This becomes clearer when considering the analogous case of GNNs with bijective activations and their equivalence with the WL test. Despite the bijectivity of the activation function, assessing separation power of these GNNs is non-trivial and has given rise to an entire branch of research over the past years.
> For clarity, in the revised version of the paper, we have clarified that injectivity or monotonicity are not impacting separability for non-polynomial activations in the comment before Theorem 2.

---

> ### Author Response · Authors · 2024-11-21
> **Response to Reviewer 1QZn - Part 2/2**
>
> **On depth stabilization:**
>
> > Besides, the set $\rho(\mathcal{N})$ is assumed to have a lower bound. As a result, when we increase the depth to infinity, it is obvious that the identified set $\rho(\mathcal{N})$ would not change after a threshold of the depth. So a more explicit result would be more interesting.
>
> The set $\rho(\mathcal{N})$ is a subspace of $\mathbb{R}^n$ for some $n$, and it is indeed straightforward to see that it is lower bounded, monotonically decreasing, and thus asymptotically stabilizing. The non-trivial claim of Theorem 3, which requires a formal proof, is that it *stabilizes in a finite number of steps*.
> Indeed, this is not always the case. For example, consider the sequence of sets $\left([0, \frac{1}{n}]\right)_{n \in \mathbb{N}}$; although it decreases as $n \to \infty$ and is lower bounded by the set $\{ 0 \}$, it does not stabilize *in a finite number of steps*.
>
> **Other aspects:**
>
> >What's more, the network with biases and input $x$ may be equivalent with that without baises and input $(x, 1)$, which may further extend your results.
>
> We thank the reviewer for the insightful observation and acknowledge that we had not considered this approach for generalizing to the case without bias. Upon further consideration, however, we believe this approach would not work in our setting, as this equivalence does not hold in the equivariant context.
>
> In particular, while any equivariant affine mapping can be represented as an equivariant linear mapping by lifting its input to a higher dimension, the reverse—which is necessary for generalizing our framework—is not always true.
>
> To see this, consider that an affine map $x\mapsto A x + b$ can always be expressed as $x \mapsto A'\begin{bmatrix}x\\\1\end{bmatrix}$, where $A'=[A | b]$. However, for a generic equivariant linear map $A'$, this decomposition may not always be feasible, as it might not decompose into $[A | b]$ such that $x \mapsto Ax + b$ remains equivariant—that is, where $A$ is equivariant and $b$ is invariant.

---

> > ### Comment · Reviewer_1QZn · 2024-11-29
> > **Response to authors**
> >
> > Thanks for your reply. After reading all reviews and your response, my concerns are addressed and I would keep my positive score.

---

### Author Response · Authors · 2024-11-21
**Response summary**

We appreciate the reviewers’ valuable comments and suggestions, which have significantly improved the quality of the manuscript.

In this general comment, we summarize the key strengths highlighted by the reviewers:

**Strengths:**
* **Overall clarity, accessibility, and soundness:** It appears that all reviewers agree the paper is well-written, with clear motivations, rigorous proofs, and transparent assumptions. Intuitive explanations and well-defined preliminaries make it accessible to a broad audience, despite its theoretical focus.
* **Problem significance:** All reviewers seem to agree that the paper addresses a significant research question. Indeed, despite extensive work on equivariant architectures, key questions remain, particularly regarding the trade-offs of imposing equivariance. To this end, the presented paper advances understanding by exploring the separation power of specific equivariant architectures.
* **Valuable insights into the factors influencing separation power:** In particular, Reviewers ESKT and rxAf seem to appreciate the new insights the paper provides into how various factors and architectural choices impact separation power.
* **Novel approach:** Reviewer rxAf found the use of the twin network trick interesting, as it transforms a network separation problem into a zero locus problem, allowing the application of functional equation theory and recursive techniques to solve zero locus problems.

We proceed to summarize the main weaknesses highlighted by the majority of reviewers, along with how we addressed them in the manuscript.

**Weaknesses:**
* **Long preliminaries:** we addressed this by streamlining the preliminary material in Section 4.1 and referring to the Appendix for a more detailed explanation. Additionally, we removed Example 1 and condensed Example 3 in Section 4.2, given its similarities with Example 2.
* **Difficulty in reading Theorem 1:** we have refactored the presentation of Theorem 1 by introducing and recalling the necessary notation prior to the theorem statement. Additionally, the theorem and its preamble now appear on the same page, enhancing readability.
* **Difficulty in understanding the impact of Theorem 1:** as suggested by reviewer ceR4, we included some proof outlines in the respective sections, specifically highlighting the role of Theorem 1 as a fundamental tool for proving nearly all subsequent results, particularly Theorems 2 and 3. To accommodate these details, we condensed the discussion of preliminaries on representation theory. If the reviewers find that these modifications hinder the manuscript quality, we are open to reverting these changes or exploring alternative solutions.

More detailed and specific weaknesses and questions will be addressed in dedicated responses to the reviewers' comments. If any of our responses are unclear, we encourage reviewers to seek further clarification. We are open and available for discussion on any aspects that could improve the manuscript.

All changes in the main text have been marked in blue in the updated PDF file to facilitate the cross reading of the revision.

---

### Author Response · Authors · 2024-11-28

With the discussion period coming to an end, we would like to express our gratitude to the reviewers for their constructive comments, thoughtful engagement in the discussion, and acknowledgment that we have addressed the issues raised in their feedback, which, in some cases, has led to improved scores.

We remain fully open to further discussion and are prepared to provide additional clarifications as needed.

---

### Meta-Review · Area_Chair_pfqw · 2024-12-20

**Metareview:**

This paper investigates the separation power of equivariant neural networks, characterizing which inputs can be distinguished by a given architecture. The authors demonstrate that factors like non-polynomial activation functions, depth, and block decomposition of hidden representations influence separation power, while adding invariant features does not. These findings contribute to a deeper theoretical understanding of equivariant networks and their expressivity.

While this work offers valuable theoretical insights into the separation power of equivariant networks, the results may be too specific to finite groups and feedforward architectures with element-wise activations to directly impact practical network design, and some reviewers found the core theorem's presentation to lack clarity. Despite these limitations, the paper presents a novel theoretical framework combining group theory and functional analysis, providing insights into how architectural choices influence the expressivity of equivariant networks. Therefore, this paper is recommended for acceptance.

**Additional Comments On Reviewer Discussion:**

Reviewers questioned the paper's impact due to its seemingly unsurprising results and limited scope focusing on specific network architectures and group representations, while also criticizing the clarity and presentation, particularly regarding the complex Theorem 1 and the lack of experimental validation. They further questioned the computational complexity of the proposed formula and the practical applicability of the theoretical findings, particularly concerning CNNs.

The authors defended the novelty of their formalization and techniques, justifying their specific setting as relevant to a significant portion of the literature on equivariant networks. They improved the paper's clarity by revising Theorem 1's presentation and adding proof sketches, acknowledged the computational limitations while emphasizing the theoretical value of their work, and explained the challenges of isolating separation power in experimental settings.

The discussion clarified a few questions and led to improvements to the presentation, but did not substantively impact the overall evaluation.

---

### Decision · Program_Chairs · 2025-01-22

Accept (Poster)